# Structure of the human dopamine transporter and mechanisms of inhibition

Dushyant Kumar Srivastava[1], Vikas Navratna[1,7], Dilip K. Tosh[2], Audrey Chinn[1], Md Fulbabu Sk[3,4,5], Emad Tajkhorshid[3,4,5], Kenneth A. Jacobson[2 ✉] & Eric Gouaux[1,6 ✉]

The neurotransmitter dopamine has central roles in mood, appetite, arousal and movement[1]. Despite its importance in brain physiology and function, and as a target for illicit and therapeutic drugs, the human dopamine transporter (hDAT) and mechanisms by which it is inhibited by small molecules and $Zn^{2+}$ are without a high-resolution structural context. Here we determine the structure of hDAT in a tripartite complex with the competitive inhibitor and cocaine analogue, (−)-2-β-carbomethoxy-3-β-(4-fluorophenyl)tropane[2] (β-CFT), the non-competitive inhibitor MRS7292[3] and $Zn^{2+}$ (ref. 4). We show how β-CFT occupies the central site, approximately halfway across the membrane, stabilizing the transporter in an outward-open conformation. MRS7292 binds to a structurally uncharacterized allosteric site, adjacent to the extracellular vestibule, sequestered underneath the extracellular loop 4 (EL4) and adjacent to transmembrane helix 1b (TM1b), acting as a wedge, precluding movement of TM1b and closure of the extracellular gate. A $Zn^{2+}$ ion further stabilizes the outward-facing conformation by coupling EL4 to EL2, TM7 and TM8, thus providing specific insights into how $Zn^{2+}$ restrains the movement of EL4 relative to EL2 and inhibits transport activity.

Dopamine and the dopaminergic circuits in the brain are intimately involved in mood, reward, motivation and movement[5]. Outside the brain, dopamine participates in signalling in the eye, cardiovascular system and pancreas[6]. Within the central nervous system, dopamine is produced by a small number of neurons located in the midbrain that project throughout the brain, acting as vehicles of dopamine release to diverse regions, including the striatum, limbic system and neocortex[7], thus explaining the profound effect of dopaminergic signalling on brain function. Dysfunction of dopaminergic signalling underpins Parkinson's disease[8] and multiple psychological disorders[9], and illicit and therapeutic drugs, including medications used to treat attention deficit hyperactivity disorder, modulate dopaminergic signal transduction[10]. Widely used therapeutic or illicit drugs, such as methylphenidate, amphetamines or cocaine, target the human dopamine transporter (hDAT), perturbing or inhibiting dopamine transport and thus disrupting dopaminergic signalling[11].

The hDAT is a member of the neurotransmitter sodium symporter (NSS) family of transporters, which in turn belong to the larger family of SLC6 transporters[12], integral membrane proteins that harness ion gradients to achieve concentrative reuptake of small molecules by way of an alternating access mechanism[13]. The hDAT uses $Na^+$ and $Cl^-$ gradients to enable substrate uptake, with $K^+$ promoting the return of the transporter to the extracellular-facing conformation, following unbinding of substrate and ions within the cytoplasm[14]. The activity of hDAT is distinct from its biogenic amine transporter relatives, however, in that transport activity is inhibited by physiologically related levels[4] of $Zn^{2+}$, which is co-released with neurotransmitters[15], as well as by synthetic small molecules, such as KM822 and MRS7292, which target largely uncharacterized, allosteric site(s)[3,16,17]. Although studies on a transport-inactive *Drosophila* dopamine transporter (dDAT) illuminated its overall structure and the mechanism by which substrates and inhibitors bind to the central site[18–20], the molecular structure of functionally active hDAT and the mechanisms of small molecules and ions acting on allosteric sites, and at the central site, remain unresolved. Here we define the binding site and non-competitive inhibition mechanism of MRS7292[3], elaborate a structure-based mechanism for $Zn^{2+}$ modulation of transport, and map the binding site of β-CFT, a high-affinity cocaine analogue.

## Overall architecture of Δ-hDAT complex

To facilitate expression and purification, we removed 56 residues from the N terminus that are predicted to be unstructured, and used the point mutant I248Y, which provided modest thermostability[21], together yielding the Δ-hDAT construct. Δ-hDAT exhibits dopamine transport (Fig. 1a) and [³H]WIN35428 binding (Fig. 1b) activities similar to the full-length, wild-type transporter[22,23]. Following expression in mammalian cells, detergent solubilization and purification in the presence of MRS7292 and β-CFT, we obtained monodisperse and homogenous Δ-hDAT (Extended Data Fig. 1a,b) for cryo-electron microscopy (cryo-EM)

[1]Vollum Institute, Oregon Health and Science University, Portland, OR, USA. [2]Molecular Recognition Section, Laboratory of Bioorganic Chemistry, National Institute of Diabetes and Digestive and Kidney Diseases, National Institutes of Health, Bethesda, MD, USA. [3]Theoretical and Computational Biophysics Group, NIH Center for Macromolecular Modeling and Visualization, Beckman Institute for Advanced Science and Technology, University of Illinois at Urbana-Champaign, Urbana, IL, USA. [4]Department of Biochemistry University of Illinois at Urbana-Champaign, Urbana, IL, USA. [5]Center for Biophysics and Quantitative Biology, University of Illinois at Urbana-Champaign, Urbana, IL, USA. [6]Howard Hughes Medical Institute, Oregon Health and Science University, Portland, OR, USA. [7]Present address: Life Sciences Institute, University of Michigan, Ann Arbor, MI, USA. ✉e-mail: kennethj@niddk.nih.gov; gouauxe@ohsu.edu

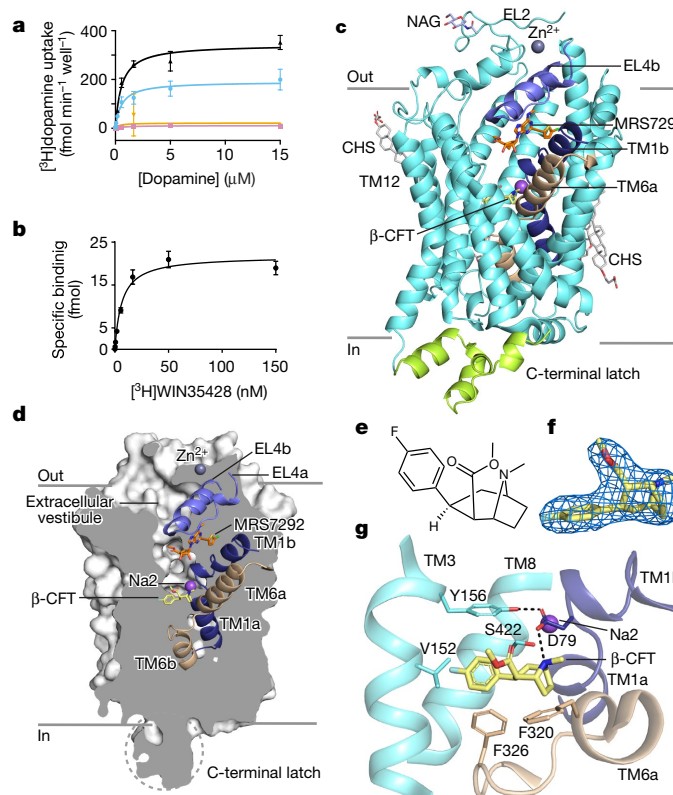

**Fig. 1 | Function and architecture of inhibitor-bound Δ-hDAT. a**, Saturation uptake of [³H]dopamine in HEK293 GnTI⁻ cells expressing Δ-hDAT (black) and full-length hDAT (blue). Uptake in the presence of 10 μM MRS7292 is shown in orange and pink for Δ-hDAT and full-length hDAT, respectively. The Michaelis constant ($K_m$) values for [³H]dopamine uptake by Δ-hDAT and full-length hDAT were 0.55 ± 0.07 and 0.56 ± 0.13 μM, and reaction rate at infinite substrate concentration ($V_{max}$) values were 342.8 ± 11.7 and 190.9 ± 10.7 fmol min⁻¹ per well, respectively. Data were analysed using a Michaelis–Menten kinetics model. The uptake assay was performed in $n = 3$ biological replicates, with each in technical triplicate. Data are mean ± s.d. **b**, Scintillation proximity assay (SPA) using [³H]WIN35428 and purified His-tagged Δ-hDAT. The dissociation constant ($K_d$) for [³H]WIN35428 binding by Δ-hDAT was 6.5 ± 0.91 nM. Data are mean ± s.d. Assays were done in $n = 3$ independent replicates, each with technical triplicates. **c**, Structure of Δ-hDAT showing β-CFT in the central site and MRS7292 in the allosteric site. NAG represents an N-acetylglucosamine modification at N188. **d**, Slab view of Δ-hDAT in a surface representation showing how the transporter adopts an outward-open conformation. **e**, Chemical structure of β-CFT (prepared using ChemDraw 18.2). **f**, Density associated with β-CFT, contoured at 10σ, within 2 Å of the ligand atoms. **g**, Close-up representation of β-CFT bound to the central site. Hydrogen-bonding interactions are shown as black, dashed lines.

grid preparation. Inclusion of both MRS7292 and β-CFT yields a highly stable complex, facilitating transporter isolation and single-particle cryo-EM studies[21]. Collection of a large single-particle cryo-EM data-set and extensive image processing (Extended Data Fig. 2a), which included ab initio-based 3D classification followed by non-uniform refinement, ultimately yielded a cryo-EM reconstruction of Δ-hDAT at 3.19 Å (Extended Data Fig. 2b–e). The resulting density map allowed for fitting of nearly the entire polypeptide chain, the placement of most side chains (Extended Data Fig. 3), the positioning of bound ligands, a Zn²⁺ and a Na⁺ ion, as well as the definition of multiple detergent or lipid molecules. Both in the single-particle classifications and in the bio-chemical analysis of the transporter, we observe detergent-solubilized Δ-hDAT as a monomer, although many previous studies find that dopamine transporters and related NSSs exist as dimers or higher-ordered

oligomers[24]. Further experiments are required to understand how to retain oligomeric species upon membrane solubilization.

The overall structure of Δ-hDAT adheres to the canonical LeuT fold[25], a conserved architecture among the SLC6 transporters, with trans-membrane helices TM1–5 related to TM6–10 by a pseudo two-fold axis of symmetry, aligned approximately parallel to the membrane (Fig. 1c). The Δ-hDAT structure resolved here, bound with multiple inhibitory small molecules and ions, adopts an outward-open confor-mation (Fig. 1d) where the central ligand binding site, also known as the S1 site, is accessible to bulk solvent via the extracellular vestibule. In accord with the outward-open conformation, the distance between two conserved residues of the extracellular gate, F320 on TM6b and Y156 on TM3, is approximately 13 Å. The cytoplasmic gate is closed, consistent with an outward-open state, with TM1a residing within the protein core (Fig. 1c), interacting extensively with TM5, TM6b and TM7. The C-terminal 'latch', which caps the cytoplasmic face of the trans-porter, is—to our knowledge—the most extensive cytoplasmic motif observed to date in an NSS (Extended Data Fig. 4a), and includes three short C-terminal helices (CT1, CT2 and CT3) that cover the cytoplasmic termini of TM3, TM10 and TM12 (Extended Data Fig. 4b), further stabi-lizing the closed conformation of the cytoplasmic gate.

The activity of hDAT and related NSSs is modulated by lipids and cholesterol[26–28] and, accordingly, we find multiple lipid or lipid-like density features that we have modelled as either linear alkyl chains or cholesteryl hemisuccinate (CHS) molecules (Extended Data Fig. 5a). We observe densities consistent with either CHS or cholesterol in the Δ-hDAT structure that are equivalent to sites in dDAT[18,19] (Extended Data Fig. 5b) and near to a cholesterol site in the human serotonin transporter[29] (hSERT) (Extended Data Fig. 5c). We also find density for CHS at an additional site, in a groove formed by TM4, TM5 and TM8 (Extended Data Fig. 5d). Covalent modification of extracellular-exposed surfaces, by way of N-linked glycosylation at N181, N188 and N205, all on EL2, confers maximal transport activity upon hDAT and, when ablated, alters the potency of cocaine-like drugs[30]. For the Δ-hDAT expressed in GnTI⁻ cells, a line that yields core N-linked carbohydrate similar to HEK293 cells[31], we observe prominent density for glycosylation at N188, whereas density for modification at N181 and N205 is too weak to model.

## Central site pharmacology

There is a high degree of amino acid sequence conservation between the monoamine transporters (MATs)—hDAT, human noradrenaline transporter (hNET) and hSERT. Nevertheless, decades of pharmacologi-cal studies have led to the development of transporter-selective small molecule inhibitors that bind to the central site. β-CFT (Fig. 1e,f), which is used in the present structure determination, has modest selectivity for binding to hDAT over hNET and hSERT[32], whereas reboxetine prefers hNET over hDAT and hSERT[33], a selectivity that is conferred by residues both within and outside of the central site[34]. The classic selective sero-tonin reuptake inhibitor S-citalopram shows a strong preference for binding to hSERT over hDAT and hNET[35]. Inspection of the complex of β-CFT with Δ-hDAT enables us to visualize key interactions between the transporter and ligand and to define the overall transporter conforma-tion, thus providing information on how residues within the binding site may sculpt transporter selectivity. The role(s) of residues outside of the central binding site in modulating ligand selectivity will require further investigation.

β-CFT occupies the central site of Δ-hDAT (Fig. 1g), consistent with its action as a competitive inhibitor of dopamine uptake[36]. Using the 'A, B and C' representation of the central site[37], the tropane moiety of β-CFT is positioned toward subsite A, facing D79 and A81 on TM1b, F76 on TM1a and G323 on TM6. The fluorophenyl moiety of β-CFT is in subsite B, near residues on TM3, TM6 and TM8. V152, S422 and Y156 par-ticipate in van der Waals contacts, and F326 on the TM6a–TM6b linker forms an edge-to-face contact with the phenyl ring of the fluorophenyl

group (Fig. 1g). Akin to the dDAT–β-CFT complex[19], the carbomethoxy group protrudes toward the base of the extracellular gate[25,38], yet does not disturb the critical hydrogen-bonding interaction between Y156 and D79[39,40]. Subsite B residues M427 and G153, when introduced into the corresponding positions in dDAT, enhance β-CFT binding[19] and, in hSERT, these same residues are leucine and alanine, respectively[29,41]. These differences in subsite B residue composition may contribute to the selectivity of Δ-hDAT for β-CFT.

Subsite C residues in MATs are involved in accommodating chemical moieties of bulky inhibitor molecules, as seen in the *S*-citalopram–hSERT complex[29], in which T497 and V501 provide a mixed polar and non-polar surface for accommodating the cyano group of *S*-citalopram (Extended Data Fig. 6a). T497 is an alanine in both hDAT and hNET; thus, differential residue composition in subsite C can explain in part the selectivity of *S*-citalopram for hSERT. The superposition of β-CFT-bound Δ-hDAT with the cocaine-bound, native, porcine SERT (pSERT) (Protein Data Bank (PDB): 8DE3[42]) and β-CFT-bound dDAT (PDB: 4XPG) structures shows overall α-carbon root mean-square deviation (r.m.s.d.) values of 1.05 and 1.08 Å, respectively, with the central site being well superimposed (Extended Data Fig. 6b). The central site in Δ-hDAT is solvent-accessible via the extracellular vestibule, as a consequence of the swung-out position of F320 (Extended Data Fig. 6c), similar to the dDAT complex with β-CFT[19]. Of note, in the complex of pSERT with the β-CFT analogue cocaine[42] (PDB: 8DE3), the side chain of the equivalent phenylalanine residue is swung-in, covering the central site and occluding the ligand from the extracellular solution (Extended Data Fig. 6c).

Molecular dynamics simulations and analysis further support the binding pose of β-CFT at the central site in the Δ-hDAT cryo-EM structure (Supplementary Fig. 2). Analysis of the fluctuation of β-CFT moieties revealed more stable tropane and fluorophenyl groups compared to the greater fluctuations and solvent exposure for the carbomethoxy group (Supplementary Fig. 3). The per-residue contact analysis (Supplementary Fig. 4) revealed that residues F76, A77, D79, A81, F320 and G323 from subsite A frequently interact with the tropane moiety, and residues V152, G153, Y156, F326, S422, A423 and G426 interact with the fluorophenyl moiety, which probably aids in stabilizing β-CFT. Additionally, consistent hydrogen bonding (Supplementary Fig. 5a) between the acidic side chain of D79 and the tropane moiety (via the tropane nitrogen atom) and comparison of each simulated binding mode with the cryo-EM model (Supplementary Fig. 6), as well as the measurement of the in silico density of spatial sampling of β-CFT (Supplementary Fig. 7) support stable binding across all simulated replicas (Extended Data Fig. 7d).

A density feature adjacent to β-CFT, and indicative of a bound ion, parallels the Na2 site in LeuT[43], dDAT[18], and hSERT[29], suggesting the presence of a similar Na$^+$-coordinating site in Δ-hDAT (Extended Data Fig. 7e). The Na$^+$ ion is coordinated with the main chain oxygens of V78 and L418 on TM1 and TM8, respectively, and side chains of D421 and S422 on TM8, with a mean coordination distance of 2.3 Å.

## MRS7292 sculpts an allosteric site

The *N*-methanocarba nucleoside analogue MRS7292 slows the unbinding of the central site ligand β-CFT[21], acts as a non-competitive inhibitor of dopamine transport, and has a chemical structure unlike previously characterized hDAT ligands[3,16]. To understand how MRS7292 inhibits dopamine transport and slows the dissociation of ligands from the central site, we determined the structure of Δ-hDAT in complex with MRS7292 and β-CFT (Fig. 2a). The density is located underneath EL4, adjacent to TM1b, and about 13 Å above the central binding site, where β-CFT is found (Fig. 2b,c). In accord with the largely non-polar character of MRS7292, the associated binding pocket is primarily hydrophobic and lined by aromatic and aliphatic amino acid residues (Fig. 2a). The MRS7292-binding site is spatially distinct from the hSERT allosteric

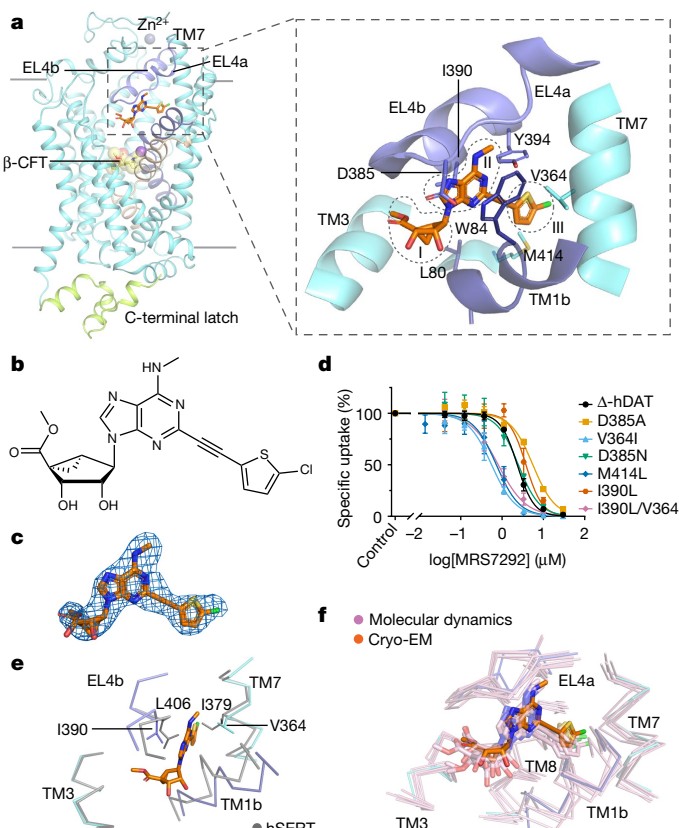

**Fig. 2 | Delineation of the MRS7292 allosteric site. a**, MRS7292 binds underneath EL4 and adjacent to TM1b, in a hydrophobic pocket additionally defined by TM7. Key residues in the MRS site are indicated in stick representation with the carbon atoms of MRS7292 in orange. Subsites I, II and III are shown. **b**, Chemical structure of MRS7292 (prepared using ChemDraw 18.2). **c**, Density associated with MRS7292, contoured at 12σ within 2 Å of the ligand atoms. **d**, Effect of MRS7292 on [³H]dopamine uptake in Δ-hDAT and Δ-hDAT mutants. Data were analysed using nonlinear regression (Methods). IC$_{50}$ measurements were performed in $n$ = 3 biological replicates (each in technical triplicate). Data are mean ± s.d. **e**, Superposition of the MRS site in Δ-hDAT and the equivalent site in hSERT (PDB: 7LIA) using α-carbon atoms of TM3 and TM8. **f**, Close-up view of the MRS site showing the cryo-EM model and the final conformations of the simulation replicas.

site (Extended Data Fig. 7a) defined by the binding of citalopram[29], vilazodone[44] or the allosterically bound serotonin molecule[41]. The allosteric site ligands in hSERT and Δ-hDAT are at least 13 Å distant from the central site, consistent with the conclusion that the allosteric ligands do not directly contact the ligand bound to the central site.

The MRS7292 compound is inserted deeply into Δ-hDAT with only a small amount of surface area exposed to the solution (Fig. 2a). Buried underneath EL4 and sandwiched between TM1b and a short helix of EL4a, the MRS7292-binding site (MRS site) can be divided into three subsites: the ring clasp (I), the adenosine sandwich (II) and the thienyl anchor (III) (Fig. 2a). The hydroxyl-decorated, rigid methanocarba ring is in subsite I, clasped underneath the turn in EL4, making extensive van der Waals interactions. The terminal alkyl group of the methyl ester is in close proximity to a hydrophobic groove formed by I159 and W162 on TM3, F391 on EL4b and F472 on TM10 (Extended Data Fig. 7b). The carbonyl oxygen of the methyl ester may interact with D476 via a water-mediated hydrogen bond, leaving the secondary hydroxyl groups as the major solvent-exposed portions of the ligand.

Gripping the MRS7292 ligand is a sandwich-like interaction between the adenine group and the indole moiety of W84 on one side, and the polypeptide main chain of residues 388–389 on the other side (Fig. 2a).

Testament to the crucial role of W84 in subsite II, mutation to alanine or cysteine severely compromises the potency of MRS7292 (Extended Data Fig. 7c,d). By contrast, the W84C mutant increases the potency of KM822, a small molecule inhibitor of hDAT that is structurally distinct from MRS7292[17], thus suggesting distinction in binding site(s) between MRS7292 and KM822. Near the indole NH group of W84, and possibly interacting with the N1 nitrogen of MRS7292 via a water molecule, is D385. Consistent with the importance of the aspartate, mutation to an alanine leads to almost 50% reduction in MRS7292-mediated inhibition of dopamine transport (Fig. 2d). Substitution of D385 by an asparagine retained potency of MRS7292 as evident by a similar inhibition curve to Δ-hDAT (Fig. 2d). This suggests that asparagine is able to form a water-mediated hydrogen bond with the N1 of MRS7292 like its aspartate counterpart in the parental Δ-hDAT. Two hydrogen bonds augment the sandwich of subsite II, one involving the exocyclic NH group of the adenine moiety and the main chain carbonyl oxygen of K384 (EL4) and the second between the hydroxyl of Y394 and the N5 atom on the adenine ring. The latter interaction is important because substitution of the N5 with a CH yields a compound that no longer inhibits hDAT transport activity[16]. The alkyne group links the adenine and thienyl moieties, in an axle-like fashion, and is surrounded by L80, V364, I390 and Y394, which together act like a bushing for the linear, carbon-carbon triple bond. In accord with the relevance of Y394, we find that substitution of Y to F diminishes the potency of MRS7292, thus substantiating the importance of the hydrogen bond between the hydroxyl group of Y394 and the N5 atom of the MRS7292 adenine ring (Extended Data Fig. 7c,d).

The 5-chlorothien-2-yl entity is deeply buried in subsite III, surrounded by a constellation of non-polar amino acids that include L280, V364, Y394, F411 and M414 (Extended Data Fig. 7e). The aromatic residues participate in edge-to-face interactions with the five-membered thienyl ring, while the aliphatic amino acids supply van der Waals contacts. M414 has a relatively flexible yet non-polar side chain, and to probe its role in the interactions with the thienyl moiety, we prepared the Δ-hDAT(M414L) variant. Notably, the M414L mutant showed an enhanced potency for MRS7292, as evidenced by an approximately threefold decrease in the half-maximal inhibitory concentration ($IC_{50}$) value compared with Δ-hDAT (Fig. 2d and Extended Data Fig. 7d). We speculate that the smaller leucine side chain creates a larger pocket to accommodate the thienyl moiety, thereby reducing steric hindrance and enhancing MRS7292-mediated inhibition of dopamine transport. Most of the residues interacting with MRS7292 are conserved in related NSSs yet the hDAT orthologues, including hNET and hSERT, are less sensitive to inhibition of uptake activity by MRS7292[19]. Of the non-conserved residues, V364 is an isoleucine in hNET and hSERT (Fig. 2e). Surprisingly, mutation of V364 to isoleucine enhanced apparent MRS7292 affinity, resulting in a nearly fourfold decrease in the $IC_{50}$ value compared with the wild-type-like parent (Fig. 2d and Extended Data Fig. 7d). To mimic a more hSERT-like MRS7292-binding pocket in Δ-hDAT, we generated a double mutant with an additional substitution of I390 to leucine in the V364I background. The double mutant showed enhanced potency towards MRS7292 with a corresponding approximately threefold decrease in $IC_{50}$ value, similar to the V364I mutant. However, I390L alone resulted in a modest increase in the $IC_{50}$ value, indicated by the marginal shift of the curve compared with Δ-hDAT. Taken together, and reminiscent of the determinants of central site ligand selectivity, the mechanism by which MRS7292 exhibits selectivity for Δ-hDAT over hNET and hSERT is partially dependent upon residues outside of the immediate binding site.

Molecular dynamics-based r.m.s.d. analysis showed stable positioning of MRS7292 across five simulation replicas (Supplementary Fig. 2), with marginal atomic fluctuations (Fig. 2f). The hydroxyl-decorated methanocarba (pseudo-ribose) moiety exhibited slightly higher fluctuations compared to the adenine and thienyl rings (Supplementary Fig. 3). The radial distribution function shows solvent exposure for the pseudo-ribose moiety of MRS7292 (Supplementary Fig. 3). Furthermore, analysis of the per-residue contact probability (Supplementary Fig. 4) showed that the 15 highest coordinating residues from Δ-hDAT stabilized MRS7292 within its cryo-EM pose. Moreover, the adenine ring of MRS7292 and the indole ring of W84 maintained a stacking-like interaction more than 99% of the time (Supplementary Table 1 and Supplementary Figs. 4 and 8). Hydrogen bond analysis (Supplementary Figs. 4 and 5) indicated stable interactions with key groups, such as the exocyclic NH group (N2 atom) of the adenine ring and the backbone carbonyl oxygen (O) of K384 (in EL4), as well as the hydroxyl group of Y394 and the N5 atom of the adenine ring. Additionally, the five-membered thienyl ring exhibited stabilization through hydrophobic interactions with L280, V363, V364, F411 and M414. Overlay of the MRS7292-binding site from all the simulation replicas with the cryo-EM pose is shown in Supplementary Fig. 9.

MRS7292 binds to Δ-hDAT by way of an induced-fit mechanism. Although we do not yet have a structure of Δ-hDAT in the absence of MRS7292, and thus cannot visualize its binding site in an apo state, by comparing the MRS site of Δ-hDAT to the equivalent region of a closely related hSERT structure (PDB: 7LIA), we speculate that the binding of MRS7292 to Δ-hDAT results in substantial conformational changes. Compared with hSERT, we estimate that MRS7292 binding displaces TM1b and TM6a by 1.9 and 2.1 Å and leads to their reorientations by 3.1 and 3.8°, respectively (Extended Data Fig. 7f). Compared with dDAT, TM1b and TM6a are displaced by 2.6 and 2.8 Å and reoriented by 9.0 and 3.2°, respectively (Extended Data Fig. 7g). We suggest that MRS7292 binding also readjusts the conformation of EL4a, as well as the turn between EL4a and EL4b, to sculpt the polypeptide chain for optimal interactions with the N-methanocarba and adenine rings of MRS7292.

In the context of MATs, the allosteric site of hSERT has been the most well characterized, beginning from when it was first suggested by ligand-unbinding studies to more recently, when it has been structurally defined in complexes of hSERT with inhibitors, including S-citalopram[29] and vilazodone[44], and with the substrate serotonin[41]. Comparison of the location of the allosteric site in hSERT to the allosteric 'MRS site' in Δ-hDAT shows that they are entirely distinct locations on extracellular-facing regions of the transporters (Extended Data Fig. 7a). Whereas the allosteric site in hSERT is largely formed by residues from TM10, TM11 and TM12, including the di-proline motif (P560-P561) in EL6 and P499 on TM10, the allosteric site in Δ-hDAT largely involves EL4 and TM1b, together with residues from TM5, TM7 and TM8. Further inspection of the region of Δ-hDAT that is equivalent to the allosteric site in hSERT provides a structural explanation for why Δ-hDAT is not sensitive to the same allosteric ligand as hSERT. As examples, structural superposition shows that P561 in hSERT is substituted by an arginine in hDAT, and the equivalent proline residue in the di-proline motif (P546) in Δ-hDAT is situated at a Cα–Cα distance of 5.9 Å from hSERT P561 (Extended Data Fig. 8a). Furthermore, P499 in hSERT is not conserved in Δ-hDAT and is substituted by T482. Similarly, the non-polar pocket formed by TM6a, TM10 and TM11 that accommodates the fluorophenyl moiety of S-citalopram in hSERT is distinct in comparison to Δ-hDAT (Extended Data Fig. 8b). The equivalent region in Δ-hDAT is more polar in nature with T316, T482, S539 in place of A331, P499 and F556, respectively, in Δ-hDAT and hSERT. Taken together, although Δ-hDAT retains the overall structural motif of the hSERT allosteric site, differences in amino acid composition mean that hSERT allosteric ligands probably do not bind to Δ-hDAT. Nevertheless, the site may still be a target for suitably tailored small molecules.

## Zinc restrains extracellular loops

$Zn^{2+}$ is packaged in vesicles and released upon vesicle fusion with the presynaptic membrane[45], modulating the activities of synaptic neurotransmitter receptors and transporters. Since the discovery of $Zn^{2+}$ inhibition of dopamine transport decades ago, several key residues

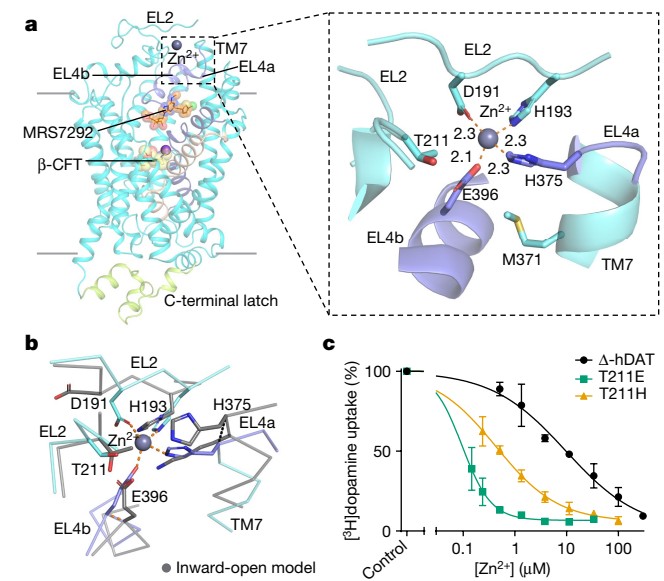

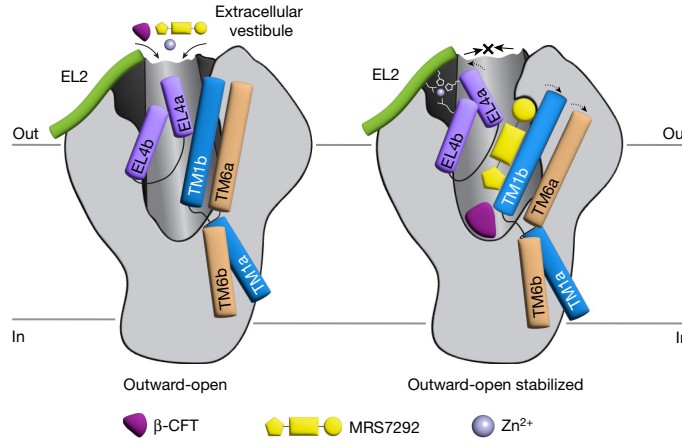

**Fig. 3 | The zinc site bridges EL2 and EL4. a,** Location and close-up view of the $Zn^{2+}$ site showing the multivalent coordination of the divalent ion. The Zn–O and Zn–N distances are expressed in Å. **b,** Superposition of the $Zn^{2+}$ site in Δ-hDAT and a predicted model of Δ-hDAT in an inward-open state showing the displacement of H375 on EL4 by 4.2 Å between the outward-open to inward-open conformations. TM3 and TM8 were aligned using α-carbon atoms. **c,** Characterization of [$^3$H]dopamine uptake in HEK293 GnTI⁻ cells expressing Δ-hDAT and the mutants T211E and T211H in the presence of $Zn^{2+}$. Data were analysed using a nonlinear regression model (Methods). The experiments were performed in *n* = 3 biological replicates with each in technical triplicate. Data are mean ± s.d.

**Fig. 4 | Mechanisms of inhibition.** β-CFT binds at the central or S1 site, stabilizing the outward-open conformation of Δ-hDAT. Binding of MRS7292 at the allosteric binding pocket near TM1b and below EL4 leads to a conformational change in TM1b, TM6 and EL4 that precludes their movement, further stabilizing the outward-open conformation and enhancing β-CFT binding at the central site. Binding of $Zn^{2+}$ restrains the EL4 loop, precluding its movement upon transition to the inward-open conformation, thus inhibiting transport.

involved in $Zn^{2+}$ binding have been identified[4]. In the absence of a high-resolution structure, proposed mechanisms of inhibition have been developed through analysis of hDAT topology, mutagenesis studies and computational modelling[4,46]. A previous study showed that $Zn^{2+}$ coordination by H193, H375 and E396 inhibits translocation of dopamine while potentiating WIN35428 binding at the central site by restraining EL2 and EL4[47]. In agreement with these findings, examination of the Δ-hDAT cryo-EM density map shows the presence of nearly continuous density between a cluster of histidine residues, a glutamate, and the aspartate D191 on EL2 and EL4, suggestive of the presence of a bound ion. Although we have not supplemented the buffers with $Zn^{2+}$ salts or ions during Δ-hDAT purification, elemental analysis of the purified Δ-hDAT protein revealed the presence of around 3.9 μM of $Zn^{2+}$, probably from the lysed cells or from the cell growth medium, the latter of which contains about 4.1 μM $Zn^{2+}$. Fitting of a $Zn^{2+}$ ion to the density yielded reasonable coordination geometry (Fig. 3a and Extended Data Fig. 9a). The $Zn^{2+}$ ion is coordinated by H193 on EL2 and H375 at the juncture of EL4a and TM7, with $Zn^{2+}$-to-nitrogen interaction distances of 2.3 Å (Fig. 3a). E396 on EL4b defines a third ligand to the $Zn^{2+}$ ion with an interaction distance of 2.1 Å (Fig. 3a). D191 was previously proposed to stabilize $Zn^{2+}$ coordination through hydrogen bonding with H193, and mutation of this residue to an asparagine resulted in an apparent threefold decrease in $Zn^{2+}$ affinity[47]; however, our cryo-EM density map shows that D191 is in close proximity to the $Zn^{2+}$ site, with a carboxylate oxygen to $Zn^{2+}$ distance of 2.3 Å (Fig. 3a). Analysis of the binding site geometry is further consistent with a bound $Zn^{2+}$ ion. A computational modelling study proposed D206 on EL2 as a fourth $Zn^{2+}$-coordinating residue[46]. However, in the current structure of Δ-hDAT, D206 is nearly 15 Å from the $Zn^{2+}$ site.

Mutation of the residues involved in $Zn^{2+}$ coordination or the introduction of the same residues at equivalent sites in related NSSs (Extended Data Fig. 9b) ablates or introduces $Zn^{2+}$ sensitivity,

respectively[4]. Thus, when H193 is mutated to lysine, the capacity for $Zn^{2+}$ to inhibit transport is compromised[4]. Similarly, when a histidine is introduced at the equivalent site in the $Zn^{2+}$-insensitive hNET, the resulting K189H variant of hNET becomes sensitive to $Zn^{2+}$ inhibition of noradrenaline transport[4]. hSERT is not sensitive to $Zn^{2+}$, probably because the residues equivalent to H193 and H375 are phenylalanine and arginine, respectively, in hSERT (Extended Data Fig. 9b).

$Zn^{2+}$ coordination in Δ-hDAT restrains EL2 in a distinct conformation compared with members of the related NSS transporters hSERT, GlyT1 and GAT1 (Extended Data Fig. 9c–e). Structures of hSERT through different states of its transport cycle have revealed that movement of EL4 relative to EL2 accompanies the transition from outward-open to inward-facing conformations[42]. By analysing the structure of Δ-hDAT in the context of the transport mechanism of hSERT, we speculate that coordination of $Zn^{2+}$ inhibits transport by restricting movement of EL4, thus preventing the conformational change from the outward-open to the inward-facing state (Fig. 3b).

Further inspection of the $Zn^{2+}$ site revealed a nearby residue, T211 (Fig. 3a), that itself is not within coordination distance of the $Zn^{2+}$ but, we hypothesized, when mutated to either a glutamate or histidine, the respective carboxyl or imidazole groups would be close enough to interact favourably with the ion (Extended Data Fig. 9f,g). To increase $Zn^{2+}$ potency in uptake experiments, we conducted assays at pH 8.5, thus favouring the deprotonated state of the coordinating histidine residues. Indeed, we found that the $IC_{50}$ values for $Zn^{2+}$ in the T211E and T211H mutants were around 100-fold and 20-fold lower, respectively, than that of Δ-hDAT (Fig. 3c). The high sensitivity of T211E to $Zn^{2+}$ made it difficult to obtain a full inhibition curve under the same conditions used for Δ-hDAT and T211H, and for this reason, a second set of $IC_{50}$ measurements was taken at pH 7.5 for the T211E mutant (Extended Data Fig. 9h). We suggest that the two mutants bind $Zn^{2+}$ with higher affinity than Δ-hDAT, thus bolstering the identification of the $Zn^{2+}$ site.

## Conclusion

Despite the overarching role of dopamine and dopaminergic signals in brain development, function and disease, and the importance of drugs in modulating the activity of hDAT, a structural understanding of transporter mechanism and allosteric inhibition has proved elusive. By elucidating the structure of Δ-hDAT bound with a trifecta

of antagonistic agents, we show how β-CFT occupies the central binding site, arresting the transporter in an outward-open conformation, adjacent to a sodium ion bound at the Na2 site. The allosteric inhibitor, MRS7292, binds above the central site and underneath EL4, immediately adjacent to TM1b, via an induced-fit mechanism, occupying a binding pocket that is not present in the closely related hSERT protein. Binding of MRS7292 displaces TM1b toward TM6a, and we speculate that the allosteric ligand locks these key helices in place, together with EL4, thus preventing isomerization of the transporter to an inward-facing state. Although mutants of residues in contact with MRS7292 reduce potency of MRS7292, swapping of non-conserved residues between hSERT and hDAT suggests that amino acids outside of immediate contact with MRS7292 also confer selective binding of MRS7292 to hDAT. A $Zn^{2+}$ ion occupies a binding site immediately above the MRS7292 ligand, coordinated by residues on both EL2 and EL4. By tethering EL4 to EL2, the bound $Zn^{2+}$ ion may restrict movement of EL4 upon transporter rearrangement to inward-facing conformations, thus providing insight into how $Zn^{2+}$ inhibits transport activity (Fig. 4). All together, these bound agents restrict the conformational mobility of Δ-hDAT, preventing isomerization to occluded or inward-facing states, and more generally, they provide fresh insights into how small molecules and ions can modulate structure and activity of MATs.

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

## Methods

### Plasmid and constructs

The hDAT cDNA (UniProt ID Q01959) was cloned into the pEG-BacMam vector[48] with an N-terminal His-StrepII–eGFP tag and a 3C protease site (LEVLFQGP) between the eGFP tag and the start of the hDAT protein-coding sequence. This construct also harboured an N-terminal deletion of the proteolytically labile 56 amino acids and included a thermostabilizing mutation (I248Y) as previously described[21] and will be referred to as Δ-hDAT throughout. Point mutations were introduced using site-directed mutagenesis verified by DNA sequencing.

### Protein expression and purification

Baculovirus-mediated expression of Δ-hDAT was performed following standard protocol[48], as previously described[21], with minor modifications. In brief, HEK293 GnTI⁻ (Ric-15) cells[31] at a density of 3.5 to $4.0 \times 10^6$ cells per ml were transduced with Δ-hDAT P2 virus at a multiplicity of infection of 2.5 and cultured in Erlenmeyer flasks at 37 °C with 8% $CO_2$ for 12 h, followed by the addition of 10 mM sodium butyrate[48]. Subsequently, the transduced cultures were shifted to 30 °C and incubated for a total of 48 h. The cells were collected by centrifugation at 4,000 rpm for 15 min (TX 1000 rotor, Thermo Scientific), washed in ice-cold phosphate buffered saline, flash frozen in liquid nitrogen, and stored at −80 °C until further use. Cell pellets were thawed on ice and resuspended in a resuspension buffer composed of 50 mM HEPES pH 7.8, 200 mM NaCl, and 20% glycerol supplemented with 0.8 μM aprotinin, 2 μg ml⁻¹ leupeptin, 2 μM pepstatin A and 1 mM phenylmethylsulfonyl fluoride (PMSF). The cells were lysed by sonication and centrifuged at 40,000 rpm for 60 min (Type 45 Ti rotor) to pellet the membrane fraction. Membrane pellets were resuspended using a Dounce homogenizer in resuspension buffer in which the glycerol concentration was raised to 30%, flash frozen, and stored at −80 °C. All centrifugation steps were carried out at 4 °C, unless otherwise stated.

Frozen membranes from 4.8 l of culture were thawed on ice and solubilized in a solution of 10 mM lauryl maltose neopentyl glycol (LMNG), 2 mM CHS, 50 mM HEPES pH 7.8, 200 mM NaCl, 10 μM MRS7292, 2 μM β-CFT, 0.8 μM aprotinin, 2 μg ml⁻¹ leupeptin, 2 μM pepstatin A, and 1 mM PMSF by constant stirring for about 3 h at 4 °C. The resulting solution was clarified by ultracentrifugation at 40,000 rpm for 60 min (Type 45 Ti rotor). Meanwhile, green fluorescent protein-nanobody (GNB) resin[49], prepared by coupling the GFP-nanobody protein to CNBr Sepharose resin at a concentration of 1 mg ml⁻¹, was equilibrated in 0.1 mM LMNG, 0.02 mM CHS, 50 mM HEPES pH 7.8, 200 mM NaCl, 25 μM palmitoyl-2-oleoyl-*sn*-glycero-3-phosphocholine (POPC) and 10% glycerol. The pre-equilibrated GNB resin was added to the solubilized membrane supernatant for binding in batch mode on a 3D shaker at 4 °C for 3 h. The protein-bound GNB resin was then packed into a gravity column and washed with a total of 12 column volumes of wash buffer consisting of 0.06% digitonin, 0.006% CHS, 50 mM HEPES pH 7.8, 200 mM NaCl, 25 μM POPC, 10% glycerol, 10 μM MRS7292, 2 μM β-CFT. The tag-free Δ-hDAT protein was eluted overnight in wash buffer containing 3C protease, concentrated, and further purified by size-exclusion chromatography (SEC) in 0.02% digitonin, 0.002% CHS, 50 mM HEPES pH 7.8, 200 mM NaCl, 4 μM MRS7292, 500 nM β-CFT and 25 μM POPC.

### Cryo-EM sample, grid preparation and data collection

SEC-purified Δ-hDAT protein was concentrated to about 7 mg ml⁻¹ using a 100 kDa cutoff filter and used immediately for preparation of cryo-EM grids. Holey grids (Quantifoil R 1.2/1.3 Au 200 mesh) were rendered hydrophilic by glow-discharge at 15 mA for 30 s and were used immediately. A solution of 3 μl of concentrated Δ-hDAT was applied to the grid and blotted for 3 s with no wait time, single blotting in 100% humidity at 15 °C, followed by plunge freezing in liquid ethane using a Vitrobot Mark IV vitrification system (Thermo Scientific). Cryo-EM data were collected using a Titan Krios (300 keV) microscope fitted with a Falcon4i direct electron detector and a Selectris X Energy filter (Thermo Scientific) and SerialEM v4.1.0 beta24 software. The images were recorded in electron event representation (EER) format using a defocus range of −1.0 to −2.5 μM, a total dose of 50 e⁻ Å⁻², a physical pixel size of 0.743 Å, and an energy filter slit width of 6 eV.

### Cryo-EM image processing

A total of 14,460 cryo-EM images in EER[50] format were imported into CryoSparc, versions 4.2.1 and 4.4.0[51] and motion corrected using patch motion correction followed by contrast transfer function (CTF) estimation and curation of the micrographs. Micrographs with poor CTF fits were discarded, leaving a total of 14,277 micrographs for further image processing. Particles were picked by reference-free blob picking using elliptical blobs with 80 Å and 150 Å minor and major axes, respectively. A total of 7,117,202 particles were picked initially. Particles were then extracted with a box size of 256 pixels, Fourier cropped to 64 pixels, and subjected to multiple rounds of 2D classification. The 2D classes showing promising transmembrane helix features were selected in three rounds of 2D classification, followed by ab initio-based 3D reconstruction and classification[52]. In brief, four ab initio classes were generated in duplicate jobs with the following parameters: initial batch size: 300; final batch size: 1,000; number of final iteration: 500; max alignment resolution: 8 Å; and initial alignment resolution: 20 Å. Particles from the best class with distinct transmembrane helices were pooled from both replicates, and duplicates were removed. Next, particles were re-extracted with the same box size as above with Fourier cropping by a factor of two. Subsequent 2D classification followed by ab initio-based 3D reconstruction–classification was repeated as described above, but with maximum and minimum alignment resolutions of 6 Å and 12 Å, respectively. The best class was then selected, and particles were pooled as previously described. Finally, the particles were re-extracted without any Fourier cropping with a box size of 384 pixels, and ab initio-based classification was carried out as previously described with initial and maximum alignment resolutions of 8 Å and 4.5 Å, respectively, and final iteration set at 350. Classes with the most well-defined Δ-hDAT features were selected for pooling particles. After removal of duplicates, the particles were used for non-uniform refinement[53], with initial low-pass resolution of 12 Å, followed by four additional passes of refinement with the minimize over per-particle scale parameter on. Non-uniform refinement resulted in a 3D reconstruction of a Δ-hDAT map at a resolution of 3.19 Å, based on a Fourier shell correlation (FSC) cutoff of 0.143 with 177,494 particles.

### Model building and refinement

The final cryo-EM map of Δ-hDAT was interpreted by fitting an AlphaFold-derived model[54] (AF-Q01959-F1) of hDAT in ChimeraX[55] using rigid body fitting. The N-terminal 56 residues were truncated in the AlphaFold model of Δ-hDAT. The fitted model and map were then manually adjusted using COOT (v0.9.8.6)[56] and then further refined in Phenix v1.20.1-4487[57] using real space refinement[58] in an iterative manner. The restraints for the MRS7292 compound were generated using the elbow program in Phenix and used in subsequent refinement steps. MolProbity[59] was used to assess the quality of the refined model with respect to geometric restraints, all atom clash score, and Ramachandran statistics, and Check My Metal was used to assess $Zn^{2+}$ and $Na^+$ site stereochemistry[60]. The comprehensive validation program in Phenix was used to obtain the final refinement statistics (Extended Data Table 1). In order to assess overfitting during refinement, the $FSC_{work}$ and $FSC_{free}$ curves were compared[61,62]. Δ-hDAT coordinates were shaken using PDB tools in Phenix with random shifts of 0.5 r.m.s.d. The resultant model was superposed, using α-carbon atoms, with the input model to confirm the change in r.m.s.d. This shaken model was refined against one of the two half-maps and the resultant model-versus-map FSC curve was termed as $FSC_{work}$.

A map-versus-model curve with this shaken-refined model and the other half-map, which was not used in any refinement, was obtained using the comprehensive cryo-EM validation tool in Phenix. This FSC curve was termed as $FSC_{free}$. The $FSC_{work}$ and $FSC_{free}$ curves were plotted and analysed for overfitting. Structural figures and illustrations were prepared using PyMOL (The PyMOL Molecular Graphics System, version 2.5.5, Schrödinger) and ChimeraX v1.6.1[55].

### Model of inward-open Δ-hDAT

The model of an inward-open conformation was generated using the SWISS model server[63] and the inward-open structure of hSERT (PDB:7LI6) as a template.

### Dopamine uptake assay

HEK293 GnTI− cells were transduced with Δ-hDAT and full-length hDAT P2 viruses, propagated in SF9 cells using standard methods as described in 'Protein expression and purification', at a cell density of $2.5 - 3.0 \times 10^6$ cells per ml, followed by incubation at 37 °C for 6 h. After 6 h, sodium butyrate was added to a final concentration of 10 mM, and the cells were transferred to 30 °C with 8% $CO_2$ for 6 h. The transduced cells were seeded into 96-well poly-D-lysine coated Isoplates (Perkin Elmer) at a density of 100,000 cells per well. The plates were then incubated at 30 °C for 12–16 h before the uptake assay was initiated. The cells were initially washed with 37 °C uptake assay buffer composed of 25 mM HEPES pH 7.4, 120 mM NaCl, 5 mM KCl, 1.2 mM $CaCl_2$, 1.2 mM $MgSO_4$, 5 mM D-glucose, 1 mM ascorbic acid, and 1 μM Ro 41-0960, followed by incubation in 50 μl of the same buffer for 10 min. Replicates with 10 μM of GBR12909 in the uptake assay buffer were used to measure background. For assessing the effect of MRS7292 on uptake, 10 μM of MRS7292 was added to the uptake buffer. Cells were then incubated with 50 μl of [³H]dopamine with a specific activity of 45.6 Ci mmol⁻¹ (hot:cold ratio of 1:100) in uptake assay buffer at a concentration range from 30 to 0.0137 μM for 10 min. The uptake reaction was stopped by adding 100 μl of chilled inhibition buffer, uptake assay buffer supplemented with 2.5 μM GBR12909. Two consecutive washes with 100 μl of inhibition buffer were carried out, followed by resuspension of cells in 100 μl of 1% Triton X-100. Finally, 100 μl of liquid scintillation cocktail was added to each well and [³H] counts were measured using a MicroBeta2 (Perkin Elmer). Background counts from three replicates were averaged and subtracted from total counts. Data were fit to the Michaelis–Menten equation to determine the kinetic parameters of dopamine uptake from three independent experiments ($n = 3$ biological replicates starting from transduction), each with triplicate measurements.

For [³H]dopamine uptake experiments on Δ-hDAT mutants, HEK293 GnTI− cells were transduced and incubated as described above. For $IC_{50}$ measurements of the MRS7292 compound, cells expressing Δ-hDAT and Δ-hDAT with W84A, W84C, D385A, D385N, V364I, Y394F, M414L, I390L and V364I/I390L mutations were washed with uptake buffer and then incubated with 25 μl of uptake buffer containing MRS7292 at concentrations ranging from 30 to 0.0411 μM (10 to 0.0137 μM for the M414L mutant) for 20 min. Replicates containing 10 μM of GBR12909 were used to measure background. To initiate uptake, 25 μl of 50 nM [³H] dopamine (35.5 Ci mmol⁻¹) was added, and the cells were incubated for 10 minutes. Uptake was quenched by adding 50 μl of ice-cold inhibition buffer. Subsequent washing steps and radioactivity measurements were carried out as previously described.

For $IC_{50}$ measurements of $Zn^{2+}$, uptake assays were carried out as described for MRS7292 with the following alterations. The uptake buffer contained EPPS at pH 8.5 in place of HEPES. Cells were washed once with uptake buffer containing 1 mM ethylenediaminetetraacetic acid (EDTA) to chelate ambient $Zn^{2+}$ from the cell growth medium, followed by second wash without EDTA. Cells were then incubated for 15 min in uptake buffers containing 0.3–300 μM added $Zn^{2+}$ for Δ-hDAT, 0.1–100 μM added $Zn^{2+}$ for T211H, and 0.03–30 μM added

$Zn^{2+}$ for T211E. Measurements of [³H]dopamine uptake in the absence of added $Zn^{2+}$ were obtained using buffer with 1 mM EDTA to chelate ambient $Zn^{2+}$. The $IC_{50}$ of $Zn^{2+}$ for the T211E mutant was also measured at pH 7.5 to obtain a more complete inhibition curve. Elemental analysis of the uptake buffer revealed ambient $Zn^{2+}$ present at about 100 nM. The estimated 'free' $Zn^{2+}$ concentrations are used in the $IC_{50}$ plots, where following the 1 mM EDTA wash we assume that there is 'zero' $Zn^{2+}$ and in the subsequent $Zn^{2+}$ concentrations we estimate that there is about 100 nM ambient $Zn^{2+}$, from the uptake buffer, in addition to the added $Zn^{2+}$ concentrations.

Specific counts were obtained by subtracting background counts (averaged from technical triplicates) from total counts. The specific uptake activity as percentages of the control was plotted against either MRS7292 or $Zn^{2+}$ concentrations using GraphPad Prism v7.05. Specific uptake activity in 1 pM MRS7292 and 1 mM EDTA was set to 100% for the MRS7292 and $Zn^{2+}$ $IC_{50}$ measurements, respectively. The data points were fitted using nonlinear regression models in GraphPad Prism v7.05: [inhibitor] versus normalized response with variable slope: $y = 100/(1 + (x^{Hill slope}/IC_{50}^{Hill slope}))$ for analysis of inhibition by MRS7292 and [inhibitor] versus response with variable slope: $y = bottom + (top - bottom)/(1 + (x^{Hill slope}/IC_{50}^{Hill slope}))$ for analysis of inhibition by $Zn^{2+}$. Data were collected from three independent experiments ($n = 3$ biological replicates starting from transduction), each performed with three technical replicates.

### Scintillation proximity assay

For SPA[64], His-tagged Δ-hDAT protein was purified as described in 'Protein expression and purification' but with Strep-tactin resin, utilizing the Twin strep affinity tag, and without β-CFT (WIN35428). The various buffer systems were unchanged. YSi-Cu SPA beads at 1 mg ml⁻¹ were added to Δ-hDAT (30 nM) in SEC buffer (0.02% digitonin, 0.002% CHS, 50 mM HEPES pH 7.8, 200 mM NaCl, 25 μM POPC and 4 μM MRS7292). [³H]-WIN35428 (82.8 Ci mmol⁻¹) in 20 mM HEPES pH 7.8 and 100 mM NaCl was used at concentration points ranging from 0 to 150 nM. For background measurement, 100 μM of GBR12909 was added to the assay buffer. Reactants were added to a 96-well isoplate, briefly mixed on a shaker at room temperature, and [³H] counts were recorded using a MicroBeta2. Data were collected from three independent experiments ($n = 3$), each performed in technical triplicate, using the same purified protein sample. Background subtracted counts were plotted and analysed by a single-site binding model via nonlinear regression analysis in GraphPad Prism v7.05.

### Molecular dynamics simulation

**Simulation setup.** The Δ-hDAT cryo-EM structure was used to prepare the simulation systems, after removing all unwanted molecular species except for the ligands (β-CFT and MRS7292) and the $Zn^{2+}$ ion. A missing disordered region (EL2) was modelled using the Schrödinger Prime module[65] (Schrödinger release 2023-2: Prime, Schrödinger), and the protein-prepared wizard[66] was used to assign the protonation states of titratable residues. All histidine residues were assigned as neutral (HID) except for His129, which was protonated (HIP). A disulfide bond was introduced between Cys180 and Cys189. The protein was internally hydrated using the DOWSER plugin[67,68] of VMD[69]. The CHARMM-GUI Membrane-Builder[70] was then used to construct the initial lipid bilayer for embedding the protein. The protein's orientation in the bilayer was derived from the Orientations of Proteins in Membranes (OPM) database[71]. Subsequently, the structure was inserted into a heterogeneous lipid bilayer, and sterically clashing lipid molecules were removed. The bilayer consisted of POPC and cholesterol (CHL) at a percentage ratio of 3:2. Slabs of 40-Å TIP3P water molecules were placed above and below the bilayer. $Na^+$ and $Cl^-$ counterions were added to neutralize the systems to a total salt concentration of 0.15 M, resulting in the entire simulation unit cell (102 Å × 102 Å × 134 Å) containing approximately 122,000 atoms. LEaP was utilized to assign force field parameters for

all the molecular species in the system. The Δ-hDAT protein, lipids, $Na^+$ and $Cl^-$ ions, and TIP3P waters were described using AMBER ff19SB[72], Lipid21[73], and monovalent ion parameters for TIP3P water[74], respectively. The $Zn^{2+}$ ion was described using the Li–Merz parameters[75] for highly charged metal ions. CUFIX corrections[76] were applied to nonbonded interactions between specific pairs of charged chemical moieties. LigPrep with the OPLS4 force field[77] was used to minimize the β-CFT and MRS7292 structures. In addition, the ionized state of these ligands was realized by Epik[78] at a pH value of $7.0 \pm 2.0$. AMBER force field 2 (GAFF2)[79,80] parameter sets were used for the ligands (β-CFT and MRS7292). A typical system setup is depicted in Supplementary Fig. 2.

**Simulation conditions.** All simulations were performed using the Amber20[81] suite and pmemd.cuda module. To eliminate bad contacts between solute and solvent water molecules in the system, energy minimization and equilibration simulations were conducted in three stages prior to the production runs. Firstly, energy minimization was performed while applying harmonic restraints on the lipid and solute heavy atoms ($k = 10$ kcal mol$^{-1}$ Å$^{-2}$). The entire system was then minimized for 10,000 steps, followed by an additional 5,000 steps of energy minimization using the Steepest Descent algorithm and the conjugate gradient method. Secondly, a two-step equilibration simulation was carried out. The system was first heated from 0 K to approximately 100 K, and then gradually to 310 K with the protein and lipid restrained over 100 ps in the NVT ensemble. Subsequently, all simulated complexes underwent 10 repeats of unconstrained NPT dynamics (5 ns, each) at 310 K and 1 atm. Finally, a 1.0 μs production simulation was conducted for each complex within the NPT ensemble at 310 K and 1 atm, using periodic boundary conditions. The temperature and pressure were maintained using a Langevin thermostat[82] and a Monte Carlo barostat[83], respectively. Electrostatic interactions were calculated with a distance cutoff of 10 Å, using the particle mesh Ewald (PME)[84] method. The SHAKE algorithm[85] was used to maintain all constraints for bonds involving hydrogens, and the time step was set to 2.0 fs. In total, we conducted five independent replicas, leading to a cumulative sampling of 5 μs (5 runs × 1 μs each) and storing 500,000 frames.

**Trajectory analysis.** For visualization and analysis, we used VMD[69] and AmberTools22[86,87] along with in-house scripts. To quantify the stacking of hDAT:W84 and MRS7292, we established a threshold for when their two rings form a stacking interaction as follows. The threshold is based on the distance between the heavy atom centres of masses (COMs) of the indole ring of W84 (atoms: CG, CD1, NE1, CE2, CZ2, CH2, CZ3, CE3 and CD2) and the adenine ring of MRS7292 (atoms: N3, C4, N1, C3, C5, N4, C6, N5 and C2). We considered MRS7292 and W84 molecules to be stacked if their COM distance was ≤5 Å and the angle between their ring normals was ≤45°. The per-residue contact profile was calculated using an in-house tcl script in VMD[69]. For each snapshot within each run, the distance between every heavy atom pair, from the ligands and the protein, respectively, was computed, and distances ≤4 Å were considered a contact. Across the entire trajectory, if a residue exhibited contacts with ligands for more than 40% of the total time, it was designated as having a stable contact.

## Cell line statement

Sf9 cells for generation of baculovirus and expression of recombinant antibody fragment are from Thermo Fisher (12659017, lot 421973). The cells were not authenticated experimentally for these studies. The cells were tested negative for Mycoplasma contamination using the CELLshipper Mycoplasma Detection Kit M-100 from Bionique.

## Reporting summary

Further information on research design is available in the Nature Portfolio Reporting Summary linked to this article.

## Data availability

The cryo-EM maps and coordinates for the Δ-hDAT structure have been deposited in the Electron Microscopy Data Bank (EMDB) under accession number EMD-43128 and in the Protein Data Bank (PDB) under accession code 8VBY. All molecular dynamics trajectories generated for this study and simulation input files have been deposited in a Zenodo repository and are freely available at https://doi.org/10.5281/zenodo.11391489 (ref. 88). Source data are provided with this paper.

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

**Acknowledgements** We thank members of the Gouaux and Baconguis laboratories for input, R. Courtney for manuscript preparation, C. Sun for advice on cryo-EM data processing, M. Ralle and S. Miller for ICPMS measurements, which were performed in the OHSU Elemental Analysis Core with partial support from NIH (S10OD028492), J. Coleman (University of Pittsburgh) for his suggestions on biochemical experiments and on the manuscript, A. Janowsky for discussions related to hDAT inhibitors, and T. Provitola for assistance with figures. We acknowledge use of the OHSU Multiscale Microscopy Core (MMC), the Pacific Northwest Cryo-EM Center (PNCC) and the cryo-EM facility at Janelia research campus. PNCC is supported by NIH grant U24GM129547 and accessed through EMSL (grid.436923.9), a DOE Office of Science User Facility sponsored by the Office of Biological and Environmental Research. The computational component of the project was supported by the National Institutes of Health (Grants R24-GM145965 and P41-GM104601 to E.T.). Molecular dynamics simulations were performed using ACCESS allocations (Grant MCA06N060 to E.T.) through the support of National Science Foundation grant number ACI-1548562. The experimental work was supported by an NIH grant to E.G. (R01 MH070039) and NIDDK Intramural grant to K.A.J. (ZIADK031127). E.G. is an investigator of the Howard Hughes Medical Institute and thanks Bernard and Jennifer LaCroute for generous support.

**Author contributions** D.K.S., V.N. and E.G. designed the project. D.T. and K.A.J. prepared the MRS compound. V.N. established constructs and protein expression conditions, D.K.S. prepared all cryo-EM samples and carried out cryo-EM analysis and structure determination. D.K.S. and A.C. prepared and expressed mutant constructs and performed ligand binding experiments and uptake assays. M.F.S. and E.T. carried out the molecular dynamics studies. D.K.S. and E.G. wrote the initial draft of the manuscript, and all authors edited the manuscript.

**Competing interests** The authors declare no competing interests.

**Additional information**
**Correspondence and requests for materials** should be addressed to Kenneth A. Jacobson or Eric Gouaux.

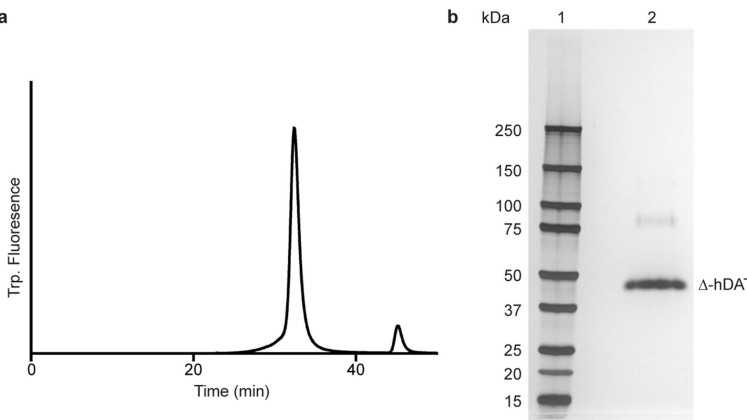

**Extended Data Fig. 1 | Biochemical characterization of Δ-hDAT.** (a) Fluorescence-detection size-exclusion chromatography (FSEC) of purified Δ-hDAT protein. (b) SDS-PAGE analysis of the FSEC sample was done once and visualized by silver staining. See Supplementary Fig. 1 for gel.

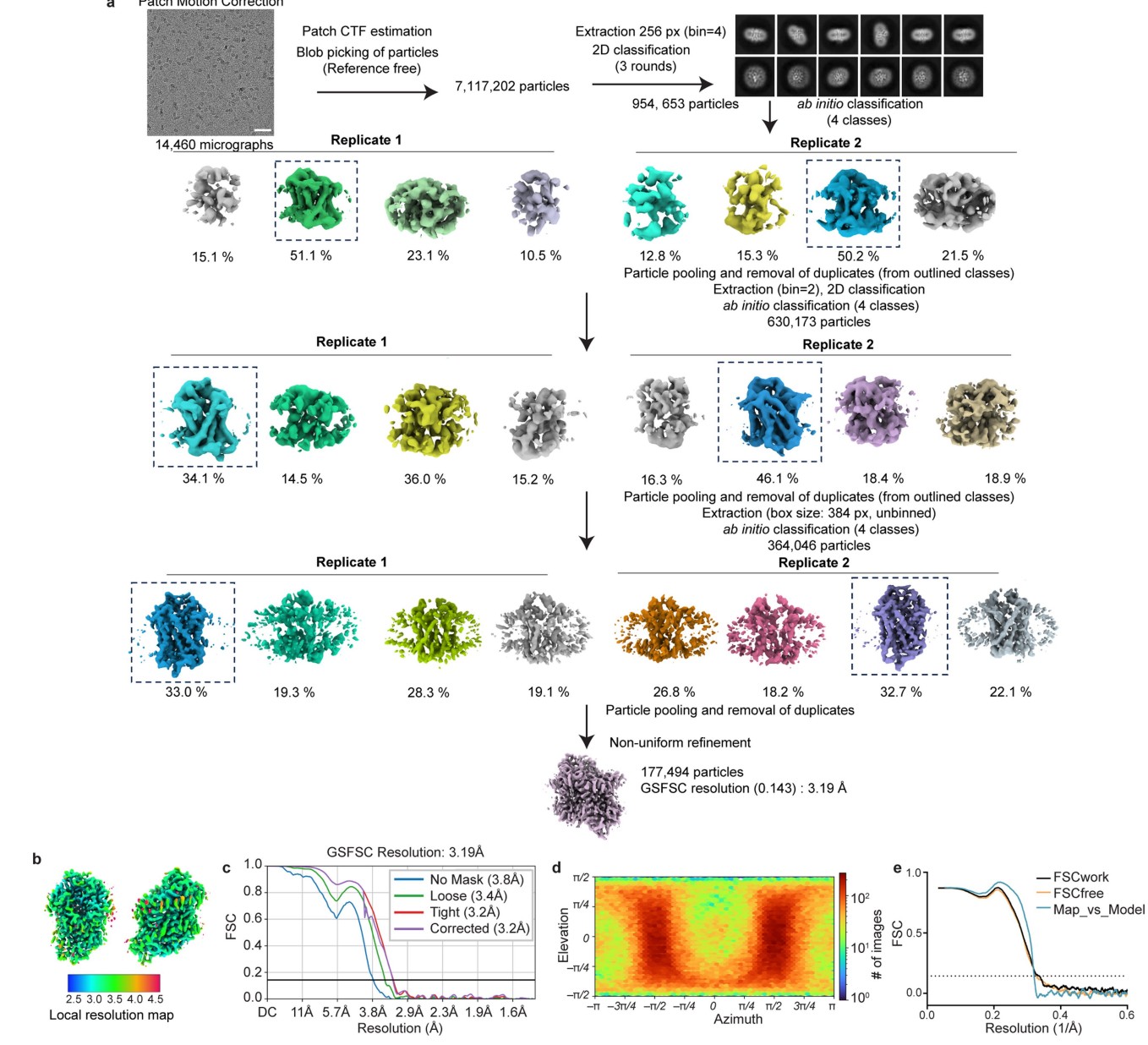

**Extended Data Fig. 2 | Cryo-EM image processing and analysis.** (a) Cryo-EM workflow. A detailed description of the image processing steps and parameters is included in the Methods section. Scale bar = 50 nm. (b) Local resolution map along with (c) the gold standard Fourier shell correlation (GSFSC) curve and (d) angular sampling of the cryo-EM reconstruction. (e) FSC$_{work}$/FSC$_{free}$ and map versus model curve.

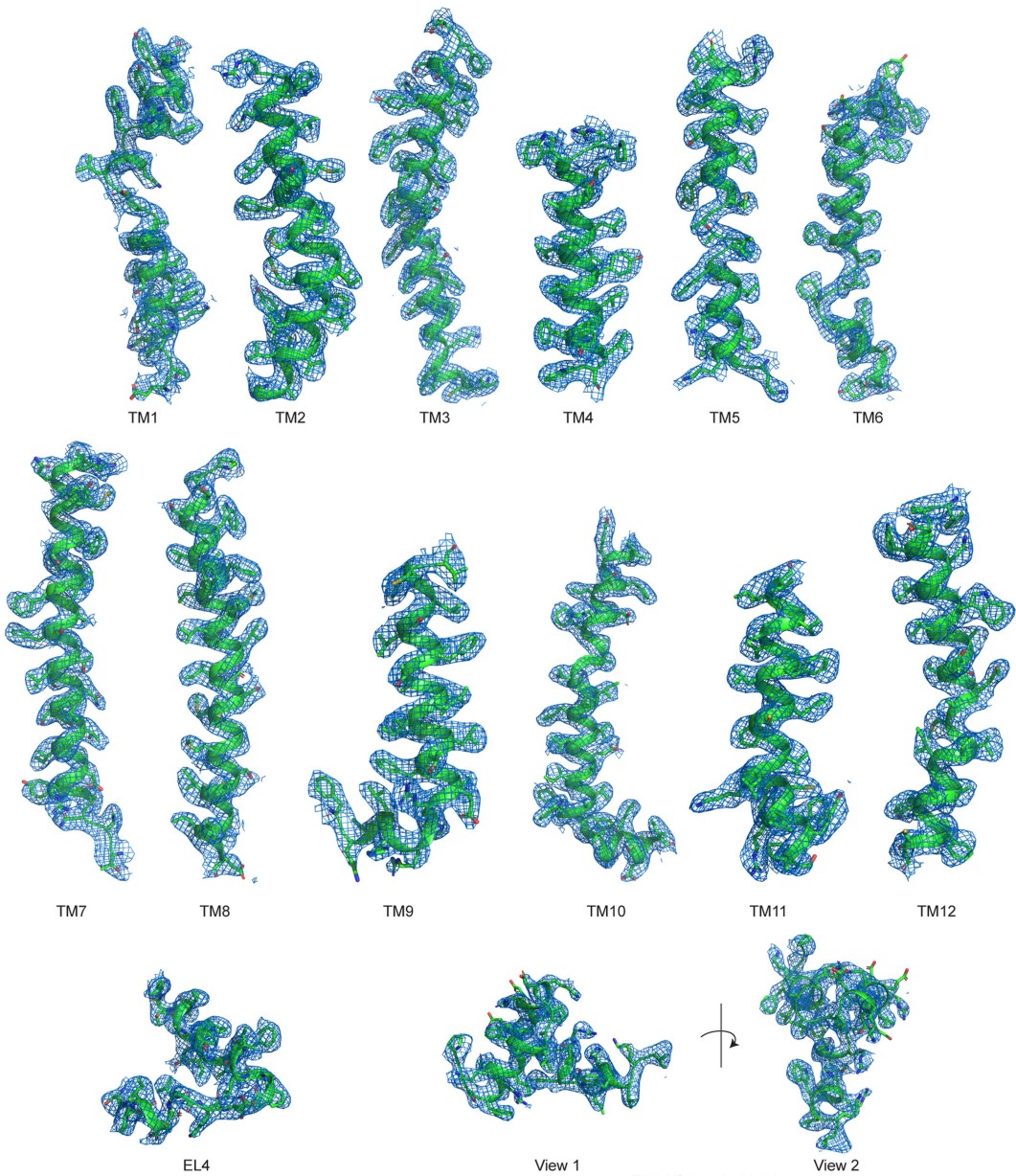

**Extended Data Fig. 3 | Density associated with transmembrane helices, EL4, and the C-terminal latch.** Isomesh map features are contoured at 8 σ and within 2 Å of the atoms associated with each feature.

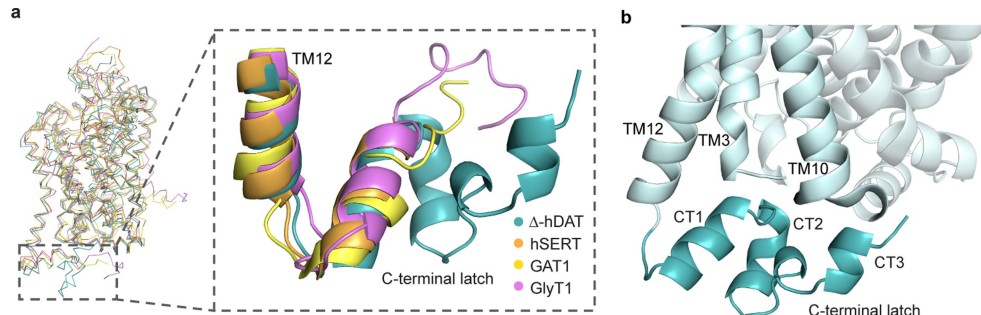

**Extended Data Fig. 4 | C-terminal latch of Δ-hDAT and structural comparison with other NSS transporters.** (a) Comparison of the C-terminal latch of Δ-hDAT with the human serotonin transporter (hSERT; PDB: 7LIA), human GABA transporter (GAT1; PDB: 7SK2), and human glycine transporter (GlyT1, PDB: 6ZBV). The structures were aligned using α-carbon atoms. (b) Close up view of the C-terminal latch of Δ-hDAT showing the proximity of TM12, TM3, and TM10.

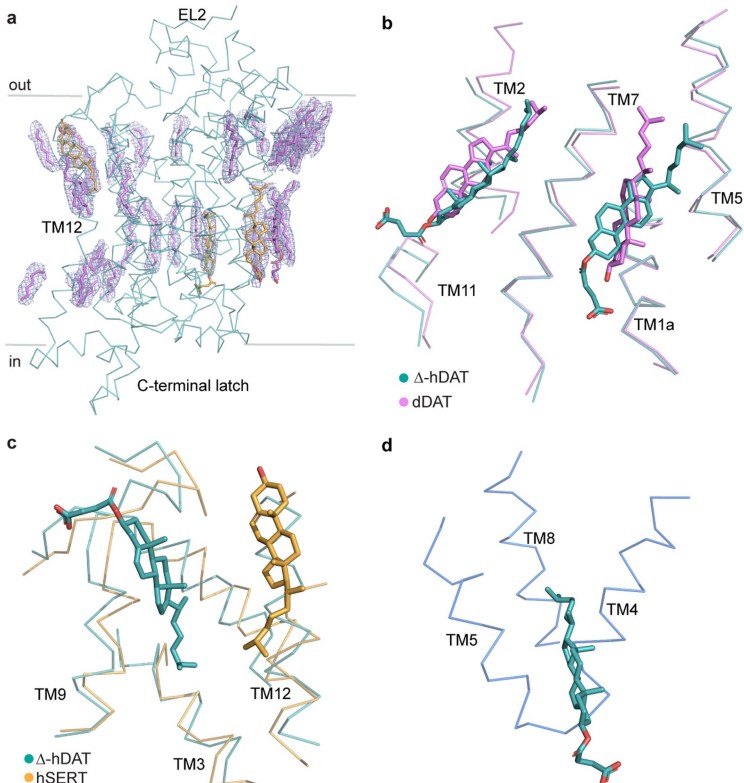

**Extended Data Fig. 5 | Lipid and lipid-like features in the Δ-hDAT reconstruction.** (a) Overall distribution of lipid-like features in the cryo-EM reconstruction of Δ-hDAT. CHS and alkyl chains of possible lipid molecules are represented as orange and magenta sticks, respectively. The associated cryo-EM density features are shown in isomesh map representation contoured at 8 σ within 2 Å of the atoms of putative CHS and alkyl chain molecules.

(b) Structural alignment of Δ-hDAT with dDAT (PDB: 4XPG) and (c) with hSERT (PDB: 5I73) showing the positions of CHS or cholesterol molecules in the proximity of lipid molecules in Δ-hDAT. Structural superposition was done using α-carbon atoms of the whole structures. (d) Additional putative CHS molecule in a groove formed by TMs 4, 5 and 8.

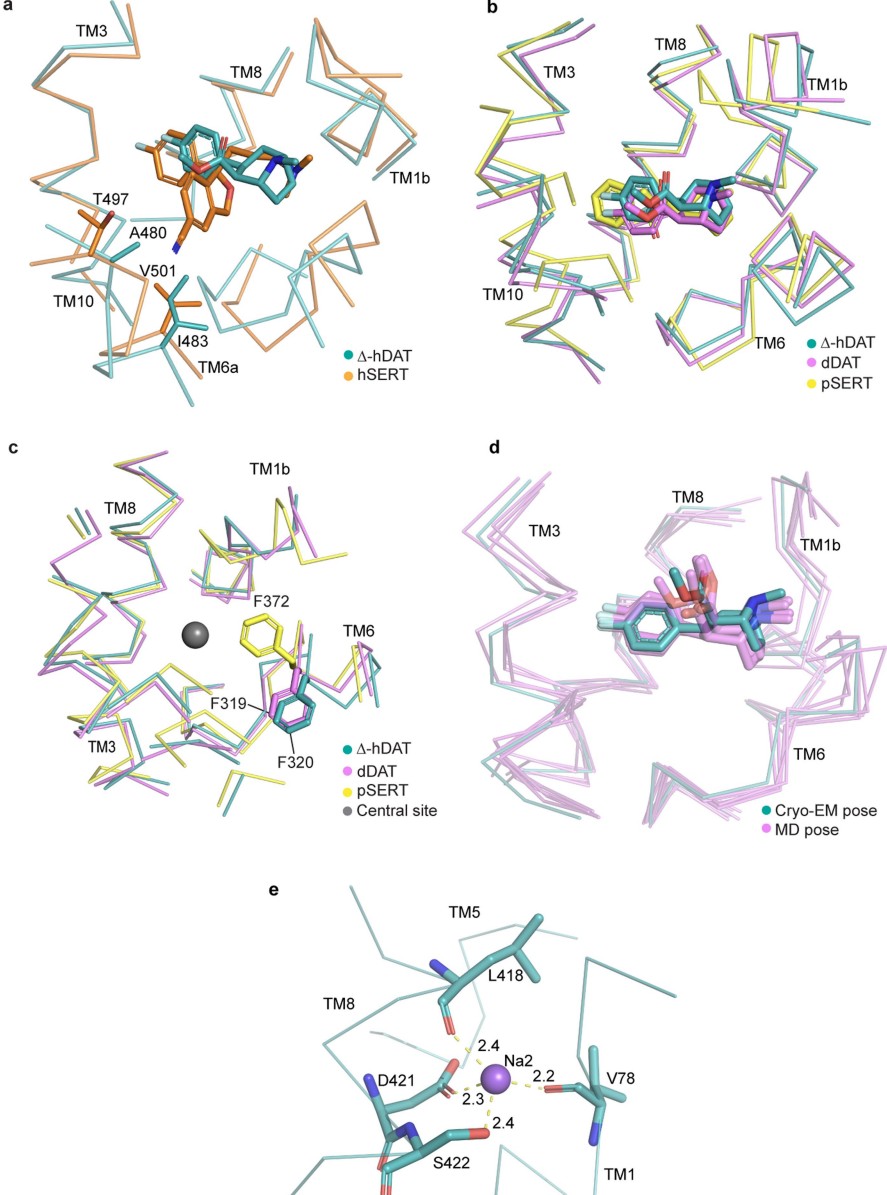

**Extended Data Fig. 6 | Central binding site of Δ-hDAT and comparison with related transporters.** (a) Overlay of the central binding pockets of S-citalopram-bound hSERT (5I73) and β-CFT-bound Δ-hDAT. The structural alignment was done using α-carbon atoms of TM3 and TM8 of Δ-hDAT. (b) Superposition of the central sites occupied by β-CFT in dDAT (PDB: 4XPG), cocaine in pSERT (PDB: 8DE3) and β-CFT in Δ-hDAT. (c) Key phenylalanines 'above' the central sites in pSERT (PDB: 8DE3) and dDAT (PDB: 4XPG) that participate in defining the occluded or outward open states. The gray sphere represents the center of mass of β-CFT in Δ-hDAT. (d) (d) Superposition of the central site showing the relative pose of β-CFT in the cryo-EM reconstruction of Δ-hDAT and five replicas of MD simulation derived poses. All structural superpositions in (b)-(d) were done using α-carbon atoms of whole structures. (e) Coordination of a Na⁺ ion at the Na2 site. The distances between the Na⁺ ion and the coordinating oxygen atoms of the likely interacting residues are shown in yellow dashed lines and represented in Å.

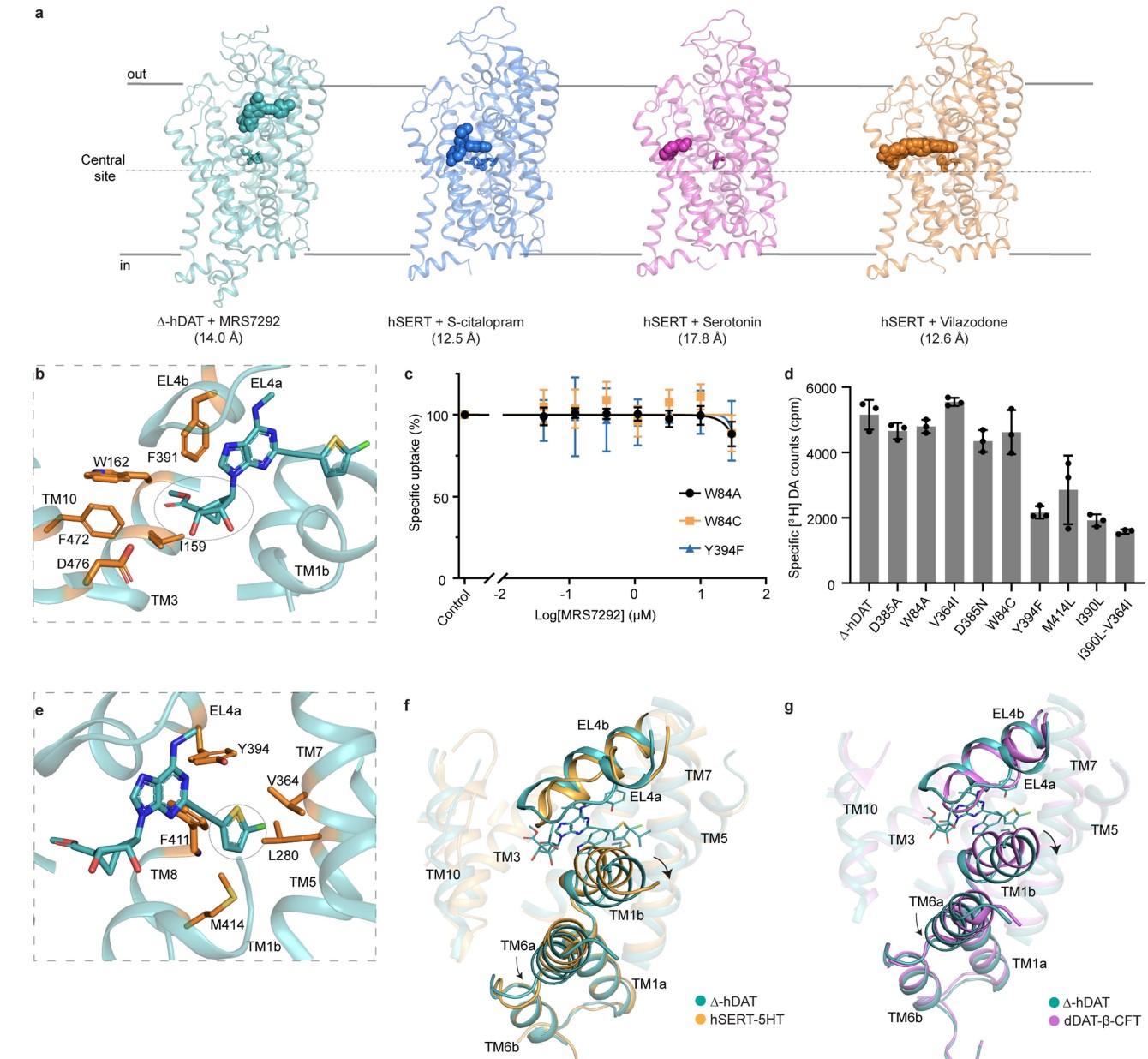

**Extended Data Fig. 7 | The MRS7292 site in Δ-hDAT and structural comparison with hSERT and dDAT.** (a) The locations of allosteric sites for the respective transporter complexes. The allosteric ligands are shown in sphere representation, and the central site molecules are shown as sticks. (b) Illustration of the MRS site showing the position of the methyl ester moiety (dashed circle). (c) Effect of MRS7292 on [³H]dopamine uptake in Δ-hDAT mutants. Data was plotted and fitted using a non-linear regression model as described in 'Methods'. Data from n = 3 biological replicates, each performed in technical triplicate, are represented as mean values +/− standard deviation. (d) Total, specific [³H]dopamine uptake for Δ-hDAT and Δ-hDAT mutant controls from the IC₅₀

experiments. Data from n = 3 biological replicates, each performed in technical triplicate, are represented as mean values +/− standard deviation. (e) Accommodation of the thienyl moiety (dashed circle) into the MRS subsite. (f) Alignment of the MRS7292 (shown as teal sticks) binding pocket with the binding pocket of hSERT in complex with serotonin (7LIA) and (g) dDAT in complex with β-CFT (4XPG). Superposition of structures was performed using α-carbon atoms of TM3 and TM8 of Δ-hDAT. The superposition shows how TMs 1b and 6a are displaced and reoriented whereas TMs 1a and 6b superimpose relatively well.

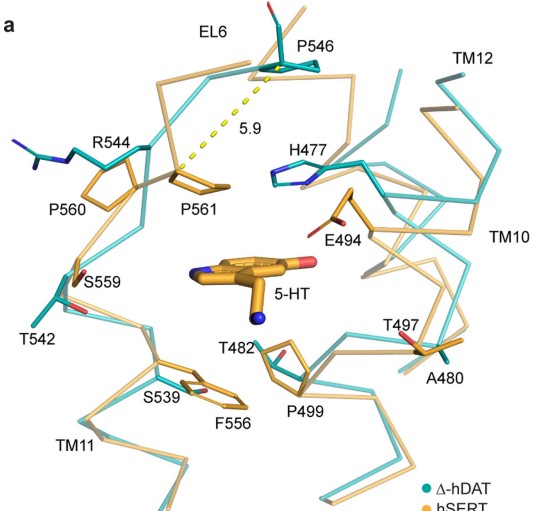
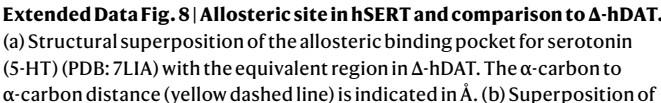

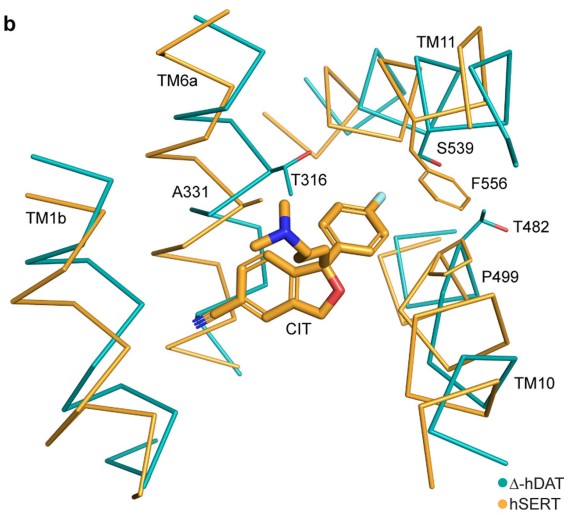

**Extended Data Fig. 8 | Allosteric site in hSERT and comparison to Δ-hDAT.** (a) Structural superposition of the allosteric binding pocket for serotonin (5-HT) (PDB: 7LIA) with the equivalent region in Δ-hDAT. The α-carbon to α-carbon distance (yellow dashed line) is indicated in Å. (b) Superposition of the S-citalopram (CIT) (PDB: 5I73) binding pocket with the equivalent region in Δ-hDAT. Superpositions were done using α-carbon atoms of TM3 and TM8. Selected residues and the allosteric molecules are represented as sticks.

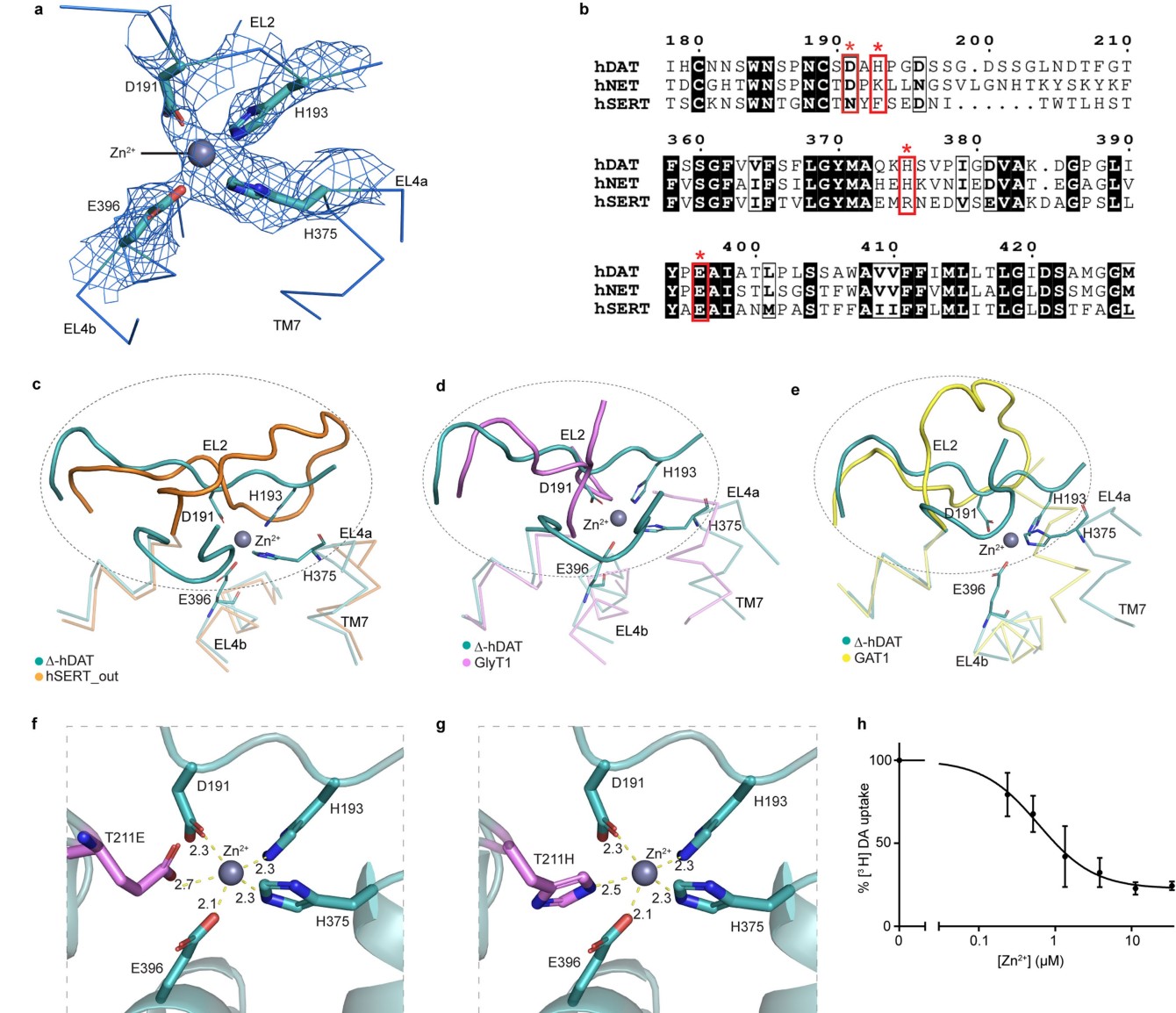

**Extended Data Fig. 9 | Zinc site analysis and alignments with other NSSs.**
(a) Isomesh map representation of coulomb density associated with the Zn²⁺ site in the Δ-hDAT cryo-EM reconstruction, contoured at 8 σ within 2.0 Å of the atoms associated with the structural feature. (b) Alignment of the amino acid sequence encompassing the Zn²⁺ binding region of Δ-hDAT with the equivalent sequences of related transporters. Residues that directly coordinate Zn²⁺ in Δ-hDAT are outlined in red. (c)-(e) Alignment of the zinc binding site of Δ-hDAT with the equivalent regions in hSERT (PDB:7LIA), the human glycine transporter (GlyT1; PDB:6ZBV), and the human GABA transporter (GAT1; PDB:7SK2)

demonstrates the unique position of EL2 in Zn²⁺-bound Δ-hDAT compared to related transporters. All structural alignments used α-carbon atoms of TM3 and TM8 of Δ-hDAT. (f,g) Simple models of T211E and T211H were created by substituting the mutated residue for T211 and selecting rotamers with favorable chi1 and chi2 angles to estimate distances to the zinc ion (Å). (h) Inhibition curve for T211E at pH 7.5. Data was analyzed using a nonlinear regression model as described in 'Methods'. Data from n = 3 biological replicates, each performed in technical triplicate, are represented as mean values +/− standard deviation.

**Extended Data Table 1 | Cryo-EM data collection and refinement statistics**

| | Δ-hDAT<br>EMD-43128, PDB 8VBY |
|---|---|
| **Data collection** | |
| Microscope | HHMI Janelia, Titan Krios |
| Electron Gun | C-FEG |
| Voltage (kV) | 300 |
| Energy filter slit width (eV) | 6 |
| Detector | Falcon4i |
| Operation mode | Counting |
| Flux on detector (e-/pix/sec) | 8.661 |
| Total electron exposure on sample (e-/Å$^2$) | 50 |
| Number of movie frames | 1096 |
| Magnification | 165K |
| Pixel size (Å) | 0.743 |
| Targeted defocus range (μm) | -1.0 to -2.5 |
| Number of collected movies | 14,460 |
| Symmetry imposed | C1 |
| Initial particle images (no.) | 7,117,202 |
| Final particle images (no.) | 177,494 |
| Map resolution (Å) FSC=0.143 | 3.19 |
| **Refinement and validation** | |
| Initial Model | AlphaFold (AF-Q01959-F1) |
| Model Resolution (Å) | |
| FSC = 0.143/0.5 | 3.1/3.3 |
| Map sharpening B factor (Å$^2$) | 128.1 |
| Model composition | |
| Non-hydrogen atoms | 4863 |
| Protein residues | 553 |
| Ligands | 41 |
| B factors (Å$^2$) | |
| Protein | 52.37 |
| Ligand | 61.16 |
| R.m.s. deviations | |
| Bond lengths (Å) | 0.003 |
| Bond angles (°) | 0.467 |
| Validation | |
| MolProbity score | 1.33 |
| Clash score | 4.37 |
| Poor rotamers (%) | 0.00 |
| Ramachandran plot | |
| Favored (%) | 97.45 |
| Allowed (%) | 2.55 |
| Disallowed (%) | 0 |

# Reporting Summary

Please do not complete any field with "not applicable" or n/a. Refer to the help text for what text to use if an item is not relevant to your study.
For final submission: please carefully check your responses for accuracy; you will not be able to make changes later.

## Statistics

For all statistical analyses, confirm that the following items are present in the figure legend, table legend, main text, or Methods section.

| n/a | Confirmed | |
|---|---|---|
| ☐ | ☑ | The exact sample size (*n*) for each experimental group/condition, given as a discrete number and unit of measurement |
| ☐ | ☑ | A statement on whether measurements were taken from distinct samples or whether the same sample was measured repeatedly |
| ☑ | ☐ | The statistical test(s) used AND whether they are one- or two-sided<br>*Only common tests should be described solely by name; describe more complex techniques in the Methods section.* |
| ☑ | ☐ | A description of all covariates tested |
| ☑ | ☐ | A description of any assumptions or corrections, such as tests of normality and adjustment for multiple comparisons |
| ☐ | ☑ | A full description of the statistical parameters including central tendency (e.g. means) or other basic estimates (e.g. regression coefficient) AND variation (e.g. standard deviation) or associated estimates of uncertainty (e.g. confidence intervals) |
| ☑ | ☐ | For null hypothesis testing, the test statistic (e.g. *F*, *t*, *r*) with confidence intervals, effect sizes, degrees of freedom and *P* value noted<br>*Give P values as exact values whenever suitable.* |
| ☑ | ☐ | For Bayesian analysis, information on the choice of priors and Markov chain Monte Carlo settings |
| ☑ | ☐ | For hierarchical and complex designs, identification of the appropriate level for tests and full reporting of outcomes |
| ☑ | ☐ | Estimates of effect sizes (e.g. Cohen's *d*, Pearson's *r*), indicating how they were calculated |

*Our web collection on statistics for biologists contains articles on many of the points above.*

## Software and code

Policy information about availability of computer code

| Data collection | SerialEM v4.1.0 beta24, MicroBeta2 Windows Workstation 2.4.0.2, Lab Solutions v5.110, VisionWorks v8.20.17096.955 |
|---|---|
| Data analysis | CryoSPARC v4.2.1 and v4.4.0, Graphpad Prism v7.05, PyMol v2.5.5, Coot v0.9.8.6, Phenix v1.20.1-4487, MolProbity (Included in PhenixPhenix v1.20.1-4487), ChimeraX v1.6.1, ChemDraw 18.2 |

For manuscripts utilizing custom algorithms or software that are central to the research but not yet described in published literature, software must be made available to editors and reviewers. We strongly encourage code deposition in a community repository (e.g. GitHub). See the Nature Portfolio guidelines for submitting code & software for further information.

## Data

Policy information about availability of data

All manuscripts must include a data availability statement. This statement should provide the following information, where applicable:
- Accession codes, unique identifiers, or web links for publicly available datasets
- A description of any restrictions on data availability
- For clinical datasets or third party data, please ensure that the statement adheres to our policy

The cryo-EM maps and coordinates for the hDAT structure have been deposited in the Electron Microscopy Data Bank (EMDB) under accession number EMD-43528 and in the Protein Data Bank under accession code 8VBY. PDB accession codes for coordinates used for comparative analysis: 7LIA, 7LI6, 4XPG, 5I73, 8DE3, 6ZBV and 7SK2.

## Research involving human participants, their data, or biological material

Policy information about studies with [human participants or human data](). See also policy information about [sex, gender (identity/presentation), and sexual orientation]() and [race, ethnicity and racism]().

| | |
|---|---|
| Reporting on sex and gender | NA |
| Reporting on race, ethnicity, or other socially relevant groupings | NA |
| Population characteristics | NA |
| Recruitment | NA |
| Ethics oversight | NA |

Note that full information on the approval of the study protocol must also be provided in the manuscript.

# Field-specific reporting

Please select the one below that is the best fit for your research. If you are not sure, read the appropriate sections before making your selection.

☒ Life sciences  ☐ Behavioural & social sciences  ☐ Ecological, evolutionary & environmental sciences

For a reference copy of the document with all sections, see [nature.com/documents/nr-reporting-summary-flat.pdf]()

# Life sciences study design

All studies must disclose on these points even when the disclosure is negative.

| | |
|---|---|
| Sample size | The maximum cryo-EM data sample size was acquired as allowed by available microscope time. |
| Data exclusions | As part of the cryo-EM image processing, exclusion of particles in order to select the best particles for 3D reconstruction is a standard and established method in the field. |
| Replication | Cryo-EM sample preparation and data acquisition and analysis were done for at least two times. Radioligand binding and uptake assays were performed in three independent replicates with each done in technical triplicate with reproducible results. |
| Randomization | No randomization was done with respect to acquisition of cryo-EM data and subsequent structure determination as it is not relevant in this kind of study. The cryo-EM data is collected automatically on selected areas of the grids in order to obtain microgr |
| Blinding | Authors were not blinded with respect to structure determination or biochemical experiments as it is not technically or practically feasible, nor important for obtaining valid results. |

# Behavioural & social sciences study design

All studies must disclose on these points even when the disclosure is negative.

| | |
|---|---|
| Study description | |
| Research sample | |
| Sampling strategy | |
| Data collection | |
| Timing | |
| Data exclusions | |
| Non-participation | |
| Randomization | |

# Ecological, evolutionary & environmental sciences study design

All studies must disclose on these points even when the disclosure is negative.

| | |
|---|---|
| Study description | |
| Research sample | |
| Sampling strategy | |
| Data collection | |
| Timing and spatial scale | |
| Data exclusions | |
| Reproducibility | |
| Randomization | |
| Blinding | |

Did the study involve field work?  ☐ Yes  ☐ No

## Field work, collection and transport

| | |
|---|---|
| Field conditions | |
| Location | |
| Access & import/export | |
| Disturbance | |

# Reporting for specific materials, systems and methods

We require information from authors about some types of materials, experimental systems and methods used in many studies. Here, indicate whether each material, system or method listed is relevant to your study. If you are not sure if a list item applies to your research, read the appropriate section before selecting a response.

## Materials & experimental systems

| n/a | Involved in the study |
|---|---|
| ☐ | ■ Antibodies |
| ☐ | ■ Eukaryotic cell lines |
| ■ | ☐ Palaeontology and archaeology |
| ■ | ☐ Animals and other organisms |
| ■ | ☐ Clinical data |
| ■ | ☐ Dual use research of concern |
| ■ | ☐ Plants |

## Methods

| n/a | Involved in the study |
|---|---|
| ■ | ☐ ChIP-seq |
| ■ | ☐ Flow cytometry |
| ■ | ☐ MRI-based neuroimaging |

## Antibodies

| | |
|---|---|
| Antibodies used | GFP Nanobody. In-house purified GFP Nanobody (Addgene #49172) was coupled to CNBr resin for affinity purification. |
| Validation | The reference for the GFP nanobody is: Kubala, M.H., O. Kovtun, K. Alexandrov, and B.M. Collins, Structural and thermodynamic analysis of the GFP: GFP-nanobody complex. Protein Sci, 2010. 19(12): p. 2389-401. |

# Eukaryotic cell lines

Policy information about cell lines and Sex and Gender in Research

| | |
|---|---|
| Cell line source(s) | Sf9 cells for generation of baculovirus and expression of recombinant antibody fragment are from Thermo Fisher (12659017, lot 421973). HEK293S GnTI- cells [Reeves P, et al. Structure and function in rhodopsin: High-level expression of rhodopsin with restricted and homogeneous N-glycosylation by a tetracycline-inducible N-acetylglucosaminyltransferase I-negative HEK293S stable mammalian cell line. Proc. Natl. Acad. Sci. USA 99 (21): 13419-13424, 2002. PubMed: 12370423] were received from Paul Reeves. |
| Authentication | The cells were not authenticated experimentally for these studies. |
| Mycoplasma contamination | The cells were tested negative for Mycoplasma contamination using the CELLshipper Mycoplasma Detection Kit M-100 from Bionique. |
| Commonly misidentified lines (See ICLAC register) | No commonly misidentified lines |

# Palaeontology and Archaeology

| | |
|---|---|
| Specimen provenance | |
| Specimen deposition | |
| Dating methods | |

☐ Tick this box to confirm that the raw and calibrated dates are available in the paper or in Supplementary Information.

| | |
|---|---|
| Ethics oversight | |

Note that full information on the approval of the study protocol must also be provided in the manuscript.

# Animals and other research organisms

Policy information about studies involving animals; ARRIVE guidelines recommended for reporting animal research, and Sex and Gender in Research

| | |
|---|---|
| Laboratory animals | |
| Wild animals | |
| Reporting on sex | |
| Field-collected samples | |
| Ethics oversight | |

Note that full information on the approval of the study protocol must also be provided in the manuscript.

# Clinical data

Policy information about clinical studies
All manuscripts should comply with the ICMJE guidelines for publication of clinical research and a completed CONSORT checklist must be included with all submissions.

| | |
|---|---|
| Clinical trial registration | |
| Study protocol | |
| Data collection | |
| Outcomes | |

# Dual use research of concern

Policy information about dual use research of concern

## Hazards

Could the accidental, deliberate or reckless misuse of agents or technologies generated in the work, or the application of information presented in the manuscript, pose a threat to:

| No | Yes | |
|----|-----|--|
| ■ | ☐ | Public health |
| ■ | ☐ | National security |
| ■ | ☐ | Crops and/or livestock |
| ■ | ☐ | Ecosystems |
| ■ | ☐ | Any other significant area |

## Experiments of concern

Does the work involve any of these experiments of concern:

| No | Yes | |
|----|-----|--|
| ■ | ☐ | Demonstrate how to render a vaccine ineffective |
| ■ | ☐ | Confer resistance to therapeutically useful antibiotics or antiviral agents |
| ■ | ☐ | Enhance the virulence of a pathogen or render a nonpathogen virulent |
| ■ | ☐ | Increase transmissibility of a pathogen |
| ■ | ☐ | Alter the host range of a pathogen |
| ■ | ☐ | Enable evasion of diagnostic/detection modalities |
| ■ | ☐ | Enable the weaponization of a biological agent or toxin |
| ■ | ☐ | Any other potentially harmful combination of experiments and agents |

# Plants

| | |
|--|--|
| Seed stocks | |
| Novel plant genotypes | |
| Authentication | |

# ChIP-seq

## Data deposition

☐ Confirm that both raw and final processed data have been deposited in a public database such as GEO.

☐ Confirm that you have deposited or provided access to graph files (e.g. BED files) for the called peaks.

| | |
|--|--|
| Data access links<br>*May remain private before publication.* | |
| Files in database submission | |
| Genome browser session<br>(e.g. UCSC) | |

## Methodology

| | |
|--|--|
| Replicates | |
| Sequencing depth | |
| Antibodies | |
| Peak calling parameters | |
| Data quality | |
| Software | |

# Flow Cytometry

## Plots

Confirm that:

☐ The axis labels state the marker and fluorochrome used (e.g. CD4-FITC).

☐ The axis scales are clearly visible. Include numbers along axes only for bottom left plot of group (a 'group' is an analysis of identical markers).

☐ All plots are contour plots with outliers or pseudocolor plots.

☐ A numerical value for number of cells or percentage (with statistics) is provided.

## Methodology

| | |
|---|---|
| Sample preparation | |
| Instrument | |
| Software | |
| Cell population abundance | |
| Gating strategy | |

☐ Tick this box to confirm that a figure exemplifying the gating strategy is provided in the Supplementary Information.

# Magnetic resonance imaging

## Experimental design

| | |
|---|---|
| Design type | |
| Design specifications | |
| Behavioral performance measures | |

| | |
|---|---|
| Imaging type(s) | |
| Field strength | |
| Sequence & imaging parameters | |
| Area of acquisition | |

Diffusion MRI ☐ Used ☐ Not used

## Preprocessing

| | |
|---|---|
| Preprocessing software | |
| Normalization | |
| Normalization template | |
| Noise and artifact removal | |
| Volume censoring | |

## Statistical modeling & inference

| | |
|---|---|
| Model type and settings | |
| Effect(s) tested | |

Specify type of analysis: ☐ Whole brain ☐ ROI-based ☐ Both

Statistic type for inference

(See Eklund et al. 2016)

Correction

## Models & analysis

| n/a | Involved in the study |
|-----|----------------------|
| ☐ ☐ | Functional and/or effective connectivity |
| ☐ ☐ | Graph analysis |
| ☐ ☐ | Multivariate modeling or predictive analysis |

Functional and/or effective connectivity

Graph analysis

Multivariate modeling and predictive analysis

