## [Peer Review File · Nature]

Manuscript Title: Structure of the human dopamine transporter and mechanisms of inhibition

Reviewer Comments & Author Rebuttals

Reviewer Reports on the Initial Version:

Referees' comments:

Referee #1 (Remarks to the Author):

Structure of the human dopamine transporter and mechanisms of allosteric inhibition

I have read the manuscript by Gouaux and coworkers with great interest. The manuscript describes fundamentally important studies on the structural and functional characterization of the human dopamine transporter (hDAT) to elucidate mechanisms of inhibition by competitive and allosteric inhibitors. The authors used cryo-electron microscopy to determine the first structure of hDAT in a complex with a competitive inhibitor β -CFT, non-competitive inhibitor MRS7292, and metal ion Zn^{2+} . The transporter is stabilized in outward-open conformation by a combination of inhibitors, and a thermostabilizing mutation. The noncompetitive inhibitor MRS7292 binds at a novel allosteric site and contributes to stabilizing the transporter's outward-open conformation. The structural observations are complemented with radioligand binding and uptake assays and shed further light on inhibitor binding sites. Even though the work described is very important, I have a few comments and concerns, which need to be addressed, as outlined below.

Major comments

1. Upon careful observation of the provided map and model, I agree that the overall map quality is certainly adequate for publication. However, the confidence with which certain molecules, and particularly the Zn ion, are modeled in a 3.7 Å map, is concerning. In particular, the Zn binding site appears to have a rather poor density, no density for individual side chains is visible, and increasing thresholds leads to one histidine and its backbone entirely disappearing. Given the modest resolution of the map, and especially given that no additional Zn was provided in the specimen, and finally the affinity for Zn being reported in the μ M range, the presence of Zn in this map is questionable at best. To me it appears as if His375 alone would fit into the remaining density rather well. Along the same lines, several cholesterol molecules appear to fit their respective densities rather poorly, and I would encourage a more conservative re-assessment of what to model and what to omit for the final model. Finally, the ligand β -CFT is very small and given the resolution of the map, I would strongly suggest MD simulations to further validate the pose of the ligand, as is typically done for structures at >3 Å resolution.

2. One of the most important questions in monoamine transporter biology is substrate selectivity. The fact that the orthosteric binding site is perfectly conserved among hNET and hDAT suggests that some allosteric effect is involved in selectivity. The authors hint into this direction at several points, but I think it would be fascinating to mutate some of the proposed residues (subsite B/C) in uptake assays, to confirm their hypothesis, rather than use purely speculative statements.
3. The third most important point is the exceedingly short “Conclusion” section of a grand total of 3 sentences. Given the importance of this transporter for human physiology and pharmacology, I think it would be imperative to include a more thorough description of the impact of this work, especially the novel allosteric site, as well as further discussion about other potential compounds, conformational states, among others. There is also no mention about the unique posttranslational modification of hDAT - palmitoylation, etc. Overall, a more expanded introduction and conclusion section would be critically important.
4. Glycosylations are discussed in the main text, and modeled in the map, while the cell line used for expression is GnTI⁻, which is an important caveat to highlight.
5. The authors showed that the Δ -hDAT mutants Y394A and V364I/I390L have opposite effects on [³H]WIN35428 binding, however, both constructs lack any dopamine uptake

activity. It would be beneficial to provide some speculations, or ideally even explanation for these observations. Also, it would be important to add the dopamine uptake data to the extended figures as a control.

6. The comparison of dDAT and Δ -hDAT displayed a local conformational change in the TM11 position. Is there any functional relevance attributed to this conformational difference or is it purely a structural accommodation for the additional ligands bound in Δ -hDAT compared to dDAT?
7. As the authors mentioned, the C-terminal latch in Δ -hDAT is the most extensive cytoplasmic motif of any NSS observed so far and the authors suggest that it might stabilize the cytoplasmic-closed conformation. Additionally, previous work also postulated that the C-terminal latch may modulate transport activity. Have the authors made any modifications to the C-terminus and observed any differences in the activity/stability of the transporter? Also, if possible, please add more details for the C-terminal latch and its potential implications for transporter function in the manuscript.

Minor comments

1. While from a structural and biophysical standpoint, it is obvious why the authors attempted two separate ligands simultaneously for structure determination, it would be important to highlight the reasoning for it in the main text.
2. A structural comparison of Δ -hDAT and hSERT suggested an induced fit mechanism of MRS7292 binding. Given that a dDAT structure is available, please also compare it to dDAT. If possible, add a brief statement about structural differences between dDAT and Δ -hDAT at the MRS7292 binding site.
3. The text interchangeably uses one letter and three letter code for amino acids (W vs. Trp). Please pick and adhere to one consistent naming convention.
4. The word ‘stitching’ was used at several points, describing Zn coordinating residues on the extracellular face of the transporter. While it is understandable what is meant with this word, it seems to be some kind of slang(?) and should be rephrased.
5. Line 61: ‘...pharmacology of hDAT is also distinct from hSERT’. Modify to ‘...pharmacology of hDAT is also distinct from human serotonin transporter (hSERT)’.

6. Line 78: ‘...homogenous Δ -hDAT (Extended Data Fig. 1a, b) for cryo-EM...’. The referenced figures should be extended data fig. 1b, c.
7. Line 129: Modify ‘TM6a/TM6B’ to ‘TM6a/TM6b’.
8. Line 178: ‘...potency of MRS7292 (Fig. 2b) while...’. The referenced figure should be Fig 2c.
9. Line 183: ‘...dopamine transport (Fig. 2b). Two hydrogen...’. The referenced figure should be Fig 2c.
10. Line 198: ‘...in hNET and hSERT (Fig. 2c). Surprisingly, mutation...’. The referenced figure should be Fig 2b.
11. Line 200: ‘...like parent (Fig. 2b). To mimic...’. The referenced figure should be Fig 2c.
12. Line 540: Concentration for aprotinin is missing.
13. Line 551: No need to mention the full name of β -CFT.
14. In the ‘Cryo-EM sample, grid preparation and data collection’ method section, the manufacturer information for the instruments is missing.
15. In the ‘Cryo-EM data processing/ Image processing’ method section, add references for various software modules of cryoSPARC. Also, mention the version for the cryoSPARC.

Referee #2 (Remarks to the Author):

The manuscript by Srivastava and colleagues reports about the cryo-EM structure of the human dopamine transporter, complexed with a competitive inhibitor, the cocaine analogue β -CFT, the non-competitive inhibitor MRS7292, and Zn^{2+} , an ion that has been shown to bind and regulate DAT. The structure of the human DAT is a “sequel” to the structure of the drosophila DAT crystal structure, published in nature more than 10 years ago, although it is certainly the long-awaited structure of the human orthologue. It is important, that we now have the human transporter in front of us – and not a transporter from invertebrates. However, it remains to be established, how different the novel structure is in comparison to the dDAT structure. This is not yet carved out sufficiently enough in the present manuscript.

Additional novelty can be drawn from the elucidation of the binding to a hitherto uncharacterized allosteric site which allows to gain information on how allostery might function in a human transporter. On the more critical side, the manuscript (i) falls short in examining the novel allosteric binding pocket to an appreciable extent and (ii) claims that the data would provide “structural understanding of how Zn^{2+} inhibits transport activity” (line 32) – which is simply not the case.

Main points:

It is important to have the new structure of the human DAT in hands. However, it would be important if not imperative to have a more rigid comparison between the new hDAT structure (experimentally determined in the current manuscript), the alphaFold prediction and the crystal structure of dDAT: It is certainly in the interest of the readers to know how large the differences between the new hDAT structure, the dDAT, and the alphaFold model is – and to which extend the predictions based on homologous hSERT structures (solved in multiple conformation) could be utilized.

One major and novel finding in this manuscript is certainly the allosteric binding pocket in hDAT. However, the pocket is not very well explored nor described, neither structurally nor experimentally. The authors should invest some effort into a more thorough description of the binding pocket from a structural point of view and maybe also speculate more about the importance of the finding. In addition, the novel and unexpected allosteric binding pocket should be more thoroughly explored experimentally to independently verify structural findings. For instance, the I390L mutant in hDAT should be examined in more detail and compared to other single-point mutants buried deeply inside the pocket.

Binding of zinc and its impact on the DAT has been recognized as early as 1993 (Richfield, E.K. (1993) Zinc modulation of drug binding, cocaine affinity states and dopamine uptake on the dopamine uptake complex. *Mol. Pharmacol.*, 43, 100–108), a subsequent paper unveiled the zinc binding at molecular detail (Reference 4 of the current manuscript), bolstered by a study utilizing computational approaches combined with in vitro experiments and microscopy (Reference 50 of the current manuscript). Hence, the sentence “, a structure-based mechanism for how Zn^{2+} reduces transport activity of hDAT has remained unresolved.” (line 244) is not entirely true as the latter two publications contain details on this. Moreover, the authors do not show anything alike a mechanistic explanation of the zinc effect on

DAT-related functionality – hence, the statement needs to be toned down to what can be said: the authors confirmed the zinc binding site to hDAT. In addition, the unique conformation of EL2 is very difficult to see in Extended Data Fig 9b-d; please adjust the figure accordingly.

Regarding the experimental zinc data (e.g. line 256): “Thus, when H193 is mutated to lysine, the capacity for Zn²⁺ to inhibit transport is compromised.” There is no reference to any of the zinc studies nor are data from the authors shown. And the data shown are difficult to interpret: The authors should attempt to at least show the high- and low-affinity inhibition of the delta-hDAT by zinc. The data shown in Figure 3 do not allow judging the effect of zinc at the truncated transporter as the authors show only three points of a concentration response curve – this should be complemented to not only show the high affinity but also include the low affinity inhibition. I will comment on the quality of the experimental data separately.

Line 248 ff.: “Although we have not supplemented the buffers with zinc salts or ions during delta-hDAT purification, elemental analysis of the purified delta-hDAT protein revealed the presence of ~3.9 μM of zinc.” – What is the source of the zinc then?

In short: The zinc binding site section shows that zinc binds and also where it binds, but it does not explain the structure-based mechanism as promised earlier. Figure 4 proposes a rudimentary model (in a sketch), but this is not of sufficient structural information to be able to propose a “structure based” model and explain “structural understanding of how Zn²⁺ inhibits transport activity” (line 32, abstract). Maybe the authors should elaborate more on the proposed mechanism of how zinc might exert its effects.

Line 72: “To facilitate expression and purification, we removed 56 residues from the N-terminus” – the authors need to show how this cleavage impacts on the delta-hDAT expression. It has been established that the truncation of the amino terminus of monoamine transporters impacts on the transition of the transporters through the secretory pathway and reduces surface expression (see for instance in Torres, PMID: 12429746).

Also other mutations of hDAT need to be examined in slightly more detail: (line 178) Mutation of the conserved W84 by W84A can lead to severe expression deficits.  Please show surface expression data of W84A and W84C.

The question is whether the changed primary structure of the transporter sequence, with its resulting changes of the secondary structure (i.e., mutational variants) has an overall impact on the tertiary and quaternary structures – which may eventually lead to functional deficits: This needs to be explored.

In this regard, it might also be worthwhile to ascertain the works of the groups of Javitch, Sorkin and others who showed that oligomeric species could be retained upon membrane solubilization (with and without cross-linking by Copper-phenanthroline or else; cf. lines 86-87).

Extended Data Figure 4. C-terminal region of Δ-hDAT:

This is an interesting data piece as it touches upon the “other” terminus of DAT: How is the C-terminal conformation consistent with the ability of binding to a PDZ-domain? See for example Bjerggaard et al., 2004 (PMID: 15295038) and Rickhag et al., 2013 (PMID: 23481388).

Experimental data quality:

It is certainly not state of the art to use $n=2$, performed in triplicate (which is a standard measure), as a small population of $n=2$ has only 1 degree of freedom for estimating the standard deviation and results in very wide confidence interval. At the very least, the authors should replicate their experiments to reach at least an n of 3 (as the authors have done it in Figure 1 panel (a)). Also, the authors should use Standard deviation (as done in Figure 1a) throughout as it allows a better judgement of the quality of the experimental data and their variability.

The figures which need to be adjusted would be Figures 1b, 2c, 3c (and “standard deviation from the mean” is not a standard term), extended data figures 7 (again “standard deviation from the mean”). For better comparability, the authors should also include the uptake data of wildtype human DAT in figure 1 a and b, and also the obtained K_m , V_{max} and K_d values in the respective figure legends.

Minor points:

Abstract (line 23): “the mechanisms by which it is inhibited by small molecules and Zn^{2+} remain unknown.” The statement is too strong and negates decades of research; it is simply an overstatement that nothing is known – that was some 60 years back, when Hertting and Axelrod started to elucidate basic mechanisms (papers in nature & science, 1960). The only thing which can be stated here is that the novel structure adds to a more precise structural understanding.

Abstract (line 32): The structure and also the manuscript fails in explaining how zinc would inhibit transport. It just shows where it binds – in the current cryo-EM snapshot. Please modify the text accordingly.

Line 63: From a pharmacological point of view, cocaine cannot be viewed as “acting with high affinity to block transport activity”. Please adjust – also, cocaine has similar affinities across monoamine transporters, on a broader scale.

Line 141: “differential residue composition in subsite C can explain,” – “can explain” is too strong a statement here, as it is a speculation. Please replace by a respective wording.

Line 143: pSERT needs to be introduced and referenced, the PDB-code is not enough; citation 47 is needed here.

Figure 3: The figure shows F213, not F214. Please correct.

Extended Data Figure 7. The MRS7292 site in Δ -hDAT and comparison with allosteric sites 846 in hSERT. Please label representative amino acid side chains in panel (a).

Referee #3 (Remarks to the Author):

Srivastava et al. have reported the structure of the human dopamine transporter (hDAT) in complex with β -CFT and the allosteric inhibitor MRS7292, using cryo-EM techniques and producing quality maps. The structure reveals β -CFT occupying the central site, with MRS7292 binding at an unexpected allosteric site, and zinc ions further stabilizing the transporter in an outward-facing conformation. While structures of DAT from other species have been solved by this group many year ago, this paper presents the resolved structure of the human DAT(hDAT). In terms of new insights, the authors have utilized cryo-

EM to elucidate a low molecular weight transporter and reported a new allosteric site in hDAT.

The authors should address the following points:

1. I suggest that the authors conduct MD simulations to validate the binding poses of the two ligands.
2. Since the authors did not obtain the structure of the allosteric inhibitor MRS7292 bound to DAT, how can they further prove that MRS7292 inhibits dopamine transport? Could MD simulations or other experiments help explain this mechanism?
3. Would there be a synergistic effect when using MRS7292 in conjunction with the β -CFT central inhibitor? Does MRS7292 exhibit dose dependency for central site ligand binding?
4. Since MRS7292 exhibits selectivity for hDAT, have the mutation experiments demonstrated how this ligand achieves high specificity for DAT as compared to hNET and hSERT?
5. There should be an FSCwork/FSCfree curve to demonstrate the absence of overfitting in model refinement. A model versus map curve should also be included.
6. Given that hDAT is not very large, could the author describe cryo-EM techniques briefly in the main text?
7. Please include the densities of both ligands in the figures in the main text.
8. Show the key residues in Extended Data Figure 7e.
9. In Extended Data Table 2, the percentage of poor rotamers is somewhat high. Please try refining the structure again.
10. Line 112-113, the glycosylation of residues on EL2 is proposed to be related with conformation mobility, it is interesting point, could the author provide more evidence to prove that?

Author Rebuttals to Initial Comments:

Referees' comments:

Referee #1 (Remarks to the Author):

Major comments

- Upon careful observation of the provided map and model, I agree that the overall map quality is
certainly adequate for publication. However, the confidence with which certain molecules, and
particularly the Zn ion, are modeled in a 3.7 Å map, is concerning. In particular, the Zn binding site
appears to have a rather poor density, no density for individual side chains is visible, and increasing
thresholds leads to one histidine and its backbone entirely disappearing. Given the modest resolution
of the map, and especially given that no additional Zn was provided in the specimen, and finally the
affinity for Zn being reported in the μM range, the presence of Zn in this map is questionable at best.
To me it appears as if His375 alone would fit into the remaining density rather well. Along the same
lines, several cholesterol molecules appear to fit their respective densities rather poorly, and I would
encourage a more conservative re-assessment of what to model and what to omit for the final model.
Finally, the ligand β -CFT is very small and given the resolution of the map, I would strongly suggest
MD simulations to further validate the pose of the ligand, as is typically done for structures at >3 Å
resolution.

We revisited image processing using the most up-to-date software and have obtained an improved
reconstruction of Δ -hDAT with a GSFSC resolution (0.143 cut-off) of 3.19 Å. In the new density map,
the protein features are overall more well resolved and in particular, the definition of the ligands,
β -CFT and MRS7292, are dramatically improved. Indeed, the present density map allows us to
unambiguously position β -CFT in the central site and the MRS7292 ligand in the allosteric site. With
respect to the zinc site, there is well resolved side chain density for H193 on EL2. Clear density features
comprising the zinc site can be seen in both the half maps (shown below).

Half_map_A (4.5 rmsd)

Half_map_B (4.5 rmsd)

Refined_map (9.0 rmsd)

The current density map also has better resolved features for lipids, lipid-like molecules and/or
detergent molecules surrounding the transporter's transmembrane domain, as well as for several
solvent molecules in the extracellular vestibule, which we have modeled as water molecules, together

with clear density for a sodium ion at the Na₂ site. We note that the coordination of the sodium ion is
in agreement with higher resolution x-ray crystallographic studies of LeuT and in accord with the
distances and geometry appropriate for a sodium ion.

In addition to a density map at substantially higher resolution, we have also proceeded with molecular
dynamics studies to investigate the pose and stability of the ligands in the central and allosteric sites.
We find that the ligands adopt stable poses, consistent with the cryo-EM defined poses. We include a
thorough description of the molecular dynamics studies in a supplement to the revised manuscript.

The updated cryo-EM map and model have been deposited in EMDB and PDB under the same accession
codes.

*- One of the most important questions in monoamine transporter biology is substrate selectivity. The*
*fact that the orthosteric binding site is perfectly conserved among hNET and hDAT suggests that*
*some allosteric effect is involved in selectivity. The authors hint into this direction at several points,*
*but I think it would be fascinating to mutate some of the proposed residues (subsite B/C) in uptake*
*assays, to confirm their hypothesis, rather than use purely speculative statements.*

We appreciate this comment and while we are tempted to carry out studies such as those proposed by
the reviewer, we note that others have already carried out extensive mutagenesis and functional
studies to probe the determinants of specificity in hNET and hDAT. One such example can be found
in the work of Kristensen and colleagues (PMID: 26503701). We suggest that the nearly exhaustive
prior work has thoroughly explored the topic of the determinants of selectivity, and we have cited
such work in the revised manuscript (line 128).

*- The third most important point is the exceedingly short "Conclusion" section of a grand total of 3*
*sentences. Given the importance of this transporter for human physiology and pharmacology, I think*
*it would be imperative to include a more thorough description of the impact of this work, especially*
*the novel allosteric site, as well as further discussion about other potential compounds,*
*conformational states, among others. There is also no mention about the unique posttranslational*
*modification of hDAT - palmitoylation, etc. Overall, a more expanded introduction and conclusion*
*section would be critically important.*

In line with the reviewer's suggestion, we have expanded the Conclusion section, keeping in mind
space constraints. We note that we see no evidence for post-translational modification other than the
N-linked glycosylation, which we refer to and discuss in the main text.

*- Glycosylations are discussed in the main text, and modeled in the map, while the cell line used for*
*expression is GnTI-, which is an important caveat to highlight.*

We appreciate this comment and will note that the cells used for expression are HEK293 GnTI- cells.
We emphasize, nevertheless, that the core glycosylation is comprised of one GlcNAc residue, which is
emanating from the Asn188 residue and is the only sugar that is resolved in the density map. This
GlcNAc residue modification is the same in GnTI- cells as in 'wild-type' HEK cells. Perhaps most
importantly, the expressed transporter is robustly functional in ligand binding and transport activity.

*- The authors showed that the Δ -hDAT mutants Y394A and V364I/I390L have opposite effects on*
*[3H]WIN35428 binding, however, both constructs lack any dopamine uptake activity. It would be*
*beneficial to provide some speculations, or ideally even explanation for these observations. Also, it*
*would be important to add the dopamine uptake data to the extended figures as a control.*

We appreciate these comments and accordingly have carried out several experiments and edits to the
manuscript. First, we have prepared additional mutants, including the more subtle Y394F variant, and
have revisited uptake experiments, carrying out more extensive studies, as explained in lines 219-222
and 235-242 of the main text, and shown in Figure 2d of the main text and 7c of the Extended Data.

Upon carefully redoing uptake experiments, we have found that while the V364I/I390L mutant shows
diminished uptake activity, it is nevertheless active in substrate uptake and robustly active with respect
to ligand binding (³H WIN35428). By contrast, the Y394A variant is neither uptake active nor active
in ligand binding. These results suggest that structural features of the central site are compromised for
the Y394A mutant but remain intact for the V364I/I390L mutant. Because Y394A was inactive, we
omitted it from IC₅₀ experiments and instead used Y394F to probe the role of Y394 in MRS7292 binding.
We also observed normal ligand binding activity by the I390L mutant. Encouraged by this, we did full
IC₅₀ measurement experiments with both the V364I/I390L and I390L mutants. In both cases the total
uptake was reduced when compared to the Δ -hDAT construct, but the mutants remained sensitive to
MRS7292-mediated dopamine uptake inhibition (lines 237-242, Fig. 2d). We previously collected
ligand binding data for I390L and I390L/V364I mutants as a proxy for activity due to reduced uptake.
Because we were able to obtain full IC₅₀ curves for I390L and I390L/V364I, we are omitting the ligand
binding data (Fig. 2d, Extended Data Fig. 7c). As requested, we have shown a panel for total uptake for
all the mutants (Extended Data Fig. 7d).

Taken together, we have more completely investigated the key interactions between the transporter
and the MRS ligand and in combination with the better resolved density map, have more extensively
explored and validated the MRS binding site. The functional studies are also complemented by the
molecular dynamics studies, as described in the main text and in the supplement.

*- The comparison of dDAT and Δ -hDAT displayed a local conformational change in the TM11*
*position. Is there any functional relevance attributed to this conformational difference or is it purely*
*a structural accommodation for the additional ligands bound in Δ -hDAT compared to dDAT?*

At this juncture we are unable to ascribe a functional relevance to the local differences in TM11, a
topic that we will address in future studies.

- As the authors mentioned, the C-terminal latch in Δ -hDAT is the most extensive cytoplasmic motif of any NSS observed so far and the authors suggest that it might stabilize the cytoplasmic-closed conformation. Additionally, previous work also postulated that the C-terminal latch may modulate transport activity. Have the authors made any modifications to the C-terminus and observed any differences in the activity/stability of the transporter? Also, if possible, please add more details for the C-terminal latch and its potential implications for transporter function in the manuscript.

In a prior study we showed that deletions to the C-terminal region resulted in reduced expression (Navratna et. al., 2018, PMID: 29965988). The importance of the C-terminus in surface targeting of hDAT has also been shown in a previous study (Bjerggaard et al., PMID: 15295038). We speculate that the C-terminal latch functions in stabilizing the conformation of the cytoplasmic gate (lines 103-107). Further studies will be necessary to fully understand the importance of this feature.

Minor comments

While from a structural and biophysical standpoint, it is obvious why the authors attempted two separate ligands simultaneously for structure determination, it would be important to highlight the reasoning for it in the main text.

As articulated in the prior work of Navratna and colleagues (PMID: 29965988), the combination of β -CFT and MRS7292 substantially stabilizes the transporter and thus provides the most robust protein complex for structural investigation, bringing along the added bonus of revealing a new allosteric ligand binding site. We have now emphasized this point in the main text at lines 81-83.

A structural comparison of Δ -hDAT and hSERT suggested an induced fit mechanism of MRS7292 binding. Given that a dDAT structure is available, please also compare it to dDAT. If possible, add a brief statement about structural differences between dDAT and Δ -hDAT at the MRS7292 binding site.

We have done a comparison of the MRS7292 pocket of Δ -hDAT to the β -CFT bound dDAT structure and have added a figure panel adjacent to the Δ -hDAT-hSERT figure panel (Extended Data Fig. 7g). We have added to the main text describing the structural differences at lines 268-270.

The text interchangeable uses one letter and three letter code for amino acids (W vs. Trp). Please pick and adhere to one consistent naming convention.

Thanks. We now use the single amino acid letter convention throughout the manuscript.

*The word 'stitching' was used at several points, describing Zn coordinating residues on the*
*extracellular face of the transporter. While it is understandable what is meant with this word, it*
*seems to be some kind of slang(?) and should be rephrased.*
We now avoid the use of 'stitching', as requested.
*Line 61: '...pharmacology of hDAT is also distinct from hSERT'. Modify to '...pharmacology of hDAT*
*is also distinct from human serotonin transporter (hSERT)'.*
We have provided the full name of hSERT at the first mention in the revised manuscript (line 113).
*Line 78: '...homogenous Δ-hDAT (Extended Data Fig. 1a, b) for cryo-EM...'. The referenced figures*
*should be extended data fig. 1b, c.*
We have referenced the correct figures in the revision.
*Line 129: Modify 'TM6a/TM6B' to 'TM6a/TM6b'.*
We have made this correction.
*Line 178: '...potency of MRS7292 (Fig. 2b) while...'. The referenced figure should be Fig 2c.*
*Line 183: '...dopamine transport (Fig. 2b). Two hydrogen...'. The referenced figure should be Fig 2c.*
*Line 198: '...in hNET and hSERT (Fig. 2c). Surprisingly, mutation...'. The referenced figure should be*
*Fig 2b.*
*Line 200: '...like parent (Fig. 2b). To mimic...'. The referenced figure should be Fig 2c.*
We have corrected the figure calls in the revised manuscript.
*Line 540: Concentration for aprotinin is missing.*
The concentration of aprotinin has been added.
*Line 551: No need to mention the full name of β-CFT.*
We have removed the full name of β-CFT.

*In the ‘Cryo-EM sample, grid preparation and data collection’ method section, the manufacturer*
*information for the instruments is missing.*

We have added the manufacturer information in the relevant method section (line 698-700).

*In the ‘Cryo-EM data processing/ Image processing’ method section, add references for various*
*software modules of cryoSPARC. Also, mention the version for the cryoSPARC.*

We have added references for the modules mentioned in the methods section (line 726) and have also
defined the version of CryoSparc used in image processing in line 705.

Referee #2 (Remarks to the Author):

Main points:

*It is important to have the new structure of the human DAT in hands. However, it would be*
*important if not imperative to have a more rigid comparison between the new hDAT structure*
*(experimentally determined in the current manuscript), the alphaFold prediction and the crystal*
*structure of dDAT: It is certainly in the interest of the readers to know how large the differences*
*between the new hDAT structure, the dDAT, and the alphaFold model is – and to which extend the*
*predictions based on homologous hSERT structures (solved in multiple conformation) could be*
*utilized.*

Superposition of Δ -hDAT cryo-EM structure using c-alpha atoms with the dDAT crystal structure
(4xpg), AlphaFold model (AF-Q01959-F1, N-term truncated for superposition) and a model of hDAT
generated using 5HT-bound outward open hSERT structure (7lia) resulted in overall RMSD values of
1.08, 0.95 and 1.13 Å respectively. There are substantial local conformational differences in TM1b,
TM6a, EL2, and EL4, likely as a result of bound ligands and the Zn²⁺ ion. The Δ -hDAT structures are
compared with hSERT and dDAT structures, and the related conformational differences in structural
elements mentioned above are analyzed and discussed in this manuscript (Extended Data Fig. 7f-g; lines
262-272).

*One major and novel finding in this manuscript is certainly the allosteric binding pocket in hDAT.*
*However, the pocket is not very well explored nor described, neither structurally nor experimentally.*
*The authors should invest some effort into a more thorough description of the binding pocket from a*
*structural point of view and maybe also speculate more about the importance of the finding. In*
*addition, the novel and unexpected allosteric binding pocket should be more thoroughly explored*
*experimentally to independently verify structural findings. For instance, the I390L mutant in hDAT*
*should be examined in more detail and compared to other single-point mutants buried deeply inside*
*the pocket.*

The revised manuscript includes a more thorough description of the MRS7292 binding pocket and its
interactions with the allosteric ligand ('MRS7292 sculpts an allosteric binding site' section) and notes
the pharmacological significance of this novel allosteric site. We have also experimentally explored
the binding pocket in more depth with MD simulations (lines 246-261 and Supplementary Data) and
mutants D385N, M414L, W84C, and Y394F (Fig. 2d and Extended Data Fig. 7c). We had previously
found that D385A resulted in a two-fold increase in the IC₅₀ of MRS7292. By further probing this
residue by mutation to Asn, we found that potency of MRS7292 was maintained by Asn, likely
because Asn maintains H-bonding with N1, an interaction that is lost by mutation to Ala (Fig 2d).
The W84C mutant behaved similarly to the W84A mutant, with severe loss of MRS7292 activity,
highlighting the importance of the pi aromatic interaction with the adenine moiety of MRS7292
(Extended Data Fig. 7c). The Y394F mutation also resulted in a severe loss of MRS7292 activity due
to the loss of the H-bond between the hydroxyl group of Tyr and N5 of MRS7292 (Extended Data Fig.
7c). Interestingly, M414L appeared to better stabilize MRS7292 binding, with this mutant resulting in
a ~3-fold decrease in IC₅₀ compared to the parent construct (Fig. 2d).

We have also explored the hSERT-like I390L, V364I, and I360L-V364I mutants in more depth with
full IC₅₀ measurements for MRS7292 (Fig. 2d). The I390L mutant demonstrated an increased IC₅₀,
while the IC₅₀ measurements for V364I and the double mutant I390L-V364I were decreased by ~4-
273 fold and ~3-fold, respectively. These results suggest that determinants of MRS7292 binding specificity
and affinity are the consequence of direct ligand-protein interactions, such as in the case of W84, but
also are the result of indirect effects involving residues that are not in direct contact with the ligand.
We describe these observations in more detail at lines 232-245.

*Binding of zinc and its impact on the DAT has been recognized as early as 1993 (Richfield, E.K. (1993)*
*Zinc modulation of drug binding, cocaine affinity states and dopamine uptake on the dopamine*
*uptake complex. Mol. Pharmacol., 43, 100–108), a subsequent paper unveiled the zinc binding at*
*molecular detail (Reference 4 of the current manuscript), bolstered by a study utilizing computational*
*approaches combined with in vitro experiments and microscopy (Reference 50 of the current*
*manuscript). Hence, the sentence “, a structure-based mechanism for how Zn²⁺ reduces transport*
*activity of hDAT has remained unresolved.” (line 244) is not entirely true as the latter two*
*publications contain details on this. Moreover, the authors do not show anything alike a mechanistic*
*explanation of the zinc effect on DAT-related functionality – hence, the statement needs to be*
*downtoned to what can be said: the authors confirmed the zinc binding site to hDAT. In addition, the*
*unique conformation of EL2 is very difficult to see in Extended Data Fig 9b-d; please adjust the figure*
*accordingly.*

Norregaard et al. (1998) identified three Zn²⁺-coordinating residues in hDAT, showed that Zn²⁺
potentiates binding of the central-site ligand WIN35428, and suggested that Zn²⁺ inhibits dopamine
transport by restraining relative movements of EL2/EL4. Stockner et al. (2013) used computational
methods and *in vitro* experiments to propose a model in which Zn²⁺ stabilizes the outward open
conformation to inhibit transport and proposed a fourth Zn²⁺-coordinating residue, D206. The
current structure of hDAT builds upon these previous findings by defining the 3D organization and

the precise location of the Zn²⁺ site. Our structure shows that D206 is too far from the Zn²⁺ site for
coordination. Further image processing has allowed us to resolve an improved 3.19 Å Δ-hDAT
reconstruction that will be included in the revised manuscript. In the updated density map, we were
able to identify D191 as a fourth coordinating residue (Fig. 3a and lines 312-316). We further
validated our identification of the Zn²⁺ site and the role of Zn²⁺ in transport inhibition by creating
gain-of-function mutations at a nearby residue, T211 (Fig. 3c). Our structure largely aligns with the
previously proposed mechanisms while more precisely showing how Zn²⁺ coordination restrains EL4
to inhibit transport activity. The revised manuscript more thoroughly credits previous work in
characterizing the Zn²⁺ site (lines 298, 300, 302) and more modestly explains how our structure and
functional experiments add to the overall understanding of transport inhibition by Zn²⁺ in the context
of this prior work (lines 302-318, 331-346).

We have also adjusted Extended Data Fig 9b-d (9c-e in revised manuscript) as suggested.

*Regarding the experimental zinc data (e.g. line 256): “Thus, when H193 is mutated to lysine, the*
*capacity for Zn²⁺ to inhibit transport is compromised.” There is no reference to any of the zinc*
*studies nor are data from the authors shown. And the data shown are difficult to interpret: The*
*authors should attempt to at least show the high- and low-affinity inhibition of the delta-hDAT by*
*zinc. The data shown in Figure 3 do not allow judging the effect of zinc at the truncated transporter*
*as the authors show only three points of a concentration response curve – this should be*
*complemented to not only show the high affinity but also include the low affinity inhibition. I will*
*comment on the quality of the experimental data separately.*

As suggested, we have added a citation to this statement in the revised manuscript (line 321-322).

We have completed more thorough uptake experiments for Δ-hDAT and the two gain of function
mutants, including full dose response curves with 7 concentrations of Zn²⁺ (Fig. 3c). These
experiments allowed us to determine IC₅₀ values for zinc and to assess the impact of our mutations on
the potency of Zn²⁺ as a transport inhibitor more thoroughly.

Previous research by Norregaard et al. (1998) has demonstrated both the high- and low-affinity
inhibition of hDAT, with the low affinity inhibition by Zn²⁺ observed at millimolar concentrations.
Norregaard and colleagues qualified this observation by pointing out that “the inhibitory effects of
Zn²⁺ in millimolar concentrations may be non-specific, caused by the non-physiological toxic
concentrations of Zn²⁺,” citing Fredrickson (*Neurobiology of Zinc*, 1989), who states that “estimates of
the physiological concentration of ionic zinc in the brain range over six orders of magnitude, from
200 pM to 300 μM” and that “high concentrations (300-1000 μM) of exogenous zinc kills neurons in
vitro.” For this reason, we did not go above 300 μM in our dose-response assays and only included
high-affinity curves. The high-affinity curves in our revised manuscript demonstrate the increased
potency of Zn²⁺ as a transport inhibitor in our gain of function mutants, thus fulfilling the purpose of
supporting our characterization of the physiologically relevant, high affinity Zn²⁺ site.

*Line 248 ff: “Although we have not supplemented the buffers with zinc salts or ions during delta-*
*hDAT purification, elemental analysis of the purified delta-hDAT protein revealed the presence of*
*~3.9 μM of zinc.” – What is the source of the zinc then?*

Elemental analysis showed that the cell growth media used contains ~4.1 μM of Zn²⁺. We also suspect
that cell lysis releases additional Zn²⁺ which may contribute to the micromolar concentrations of Zn²⁺
found in our purified protein solution (see lines 305-308).

*In short: The zinc binding site section shows that zinc binds and also where it binds, but it does not*
*explain the structure-based mechanism as promised earlier. Figure 4 proposes a rudimentary model*
*(in a sketch), but this is not of sufficient structural information to be able to propose a “structure*
*based” model and explain “structural understanding of how Zn²⁺ inhibits transport activity” (line 32,*
*abstract). Maybe the authors should elaborate more on the proposed mechanism of how zinc might*
*exert its effects.*

We have elaborated on our proposed mechanism of transport inhibition by Zn²⁺ in the revised
manuscript. Figure 3b compares the current structure with a model of the inward facing
conformation of hDAT to show how EL4 is restrained by Zn²⁺ coordination, and lines 329-334 detail
a proposed mechanism in which this restraint of EL4 in proximity EL2 stabilizes the outward-open
conformation and inhibits transport.

*Line 72: “To facilitate expression and purification, we removed 56 residues from the N-terminus” –*
*the authors need to show how this cleavage impacts on the delta-hDAT expression. It has been*
*established that the truncation of the amino terminus of monoamine transporters impacts on the*
*transition of the transporters through the secretory pathway and reduces surface expression (see for*
*instance in Torres, PMID: 12429746).*

The deletion of 56 residues from the N-terminus in conjunction with the I248Y variant yields a
modestly thermostable construct which retains ligand binding properties and superior biochemical
behavior (Navratna et al., 2018). In our kinetic uptake experiments utilizing intact cells, Δ-hDAT is
fully active with comparable Km and Vmax parameters to full length hDAT (Fig 1a), which
demonstrates that a substantial fraction of the expressed protein is translocated to the cell surface.

*Also other mutations of hDAT need to be examined in slightly more detail: (line 178) Mutation of the*
*conserved W84 by W84A can lead to severe expression deficits.  Please show surface expression*
*data of W84A and W84C.*

The mutants W84A and W84C display substantial activity in transporting dopamine, in our intact
cell-based assay (Extended Data Fig. 7c-d), which demonstrates that a significant fraction of the
transporter protein is translocated to the plasma membrane. Given the measurable transport activity

of these mutants, we do not believe that experiments involving measurement of surface expression
are warranted.

*The question is whether the changed primary structure of the transporter sequence, with its resulting*
*changes of the secondary structure (i.e., mutational variants) has an overall impact on the tertiary and*
*quartary structures – which may eventually lead to functional deficits: This needs to be explored.*

The modifications of the primary sequence are either in regions of the protein that are predicted to be
unstructured, such as at the termini, or they are point mutants in or around the MRS7292 binding
site. As we have shown for all of these variants, the transporter retains activity, thus demonstrating
that it adopts a folded and functional conformation (Fig. 2d and Extended Data Fig. 7c-d). As
discussed below, we have explored the oligomerization of hDAT to a great extent, in unpublished
studies, and have found no evidence for higher ordered oligomers. If there are such species,
elaboration of their structures and properties is beyond the scope of the present work.

*In this regard, it might also be worthwhile to ascertain the works of the groups of Javitch, Sorkin and*
*others who showed that oligomeric species could be retained upon membrane solubilization (with*
*and without cross-linking by Copper-phenanthroline or else; cf. lines 86-87).*

We have explored this topic extensively, in so-far unpublished work, using conditions and
compounds (see for example studies on AIM100, PMID: 31228486) that have been proposed to
stabilize oligomeric states. We have been unable to isolate and characterize oligomers.

*Extended Data Figure 4. C-terminal region of Δ -hDAT:*

*This is an interesting data piece as it touches upon the “other” terminus of DAT: How is the C-*
*terminal conformation consistent with the ability of binding to a PDZ-domain? See for example*
*Bjerggaard et al., 2004 (PMID: 15295038) and Rickhag et al., 2013 (PMID: 23481388).*

The PDZ domains are predicted to interact at the distal c-terminus to the PDZ binding motif. We
have not explored this aspect of hDAT C-terminal binding to a PDZ-domain as part of this study.
These are experiments which can be explored further in a separate study.

Experimental data quality:

*It is certainly not state of the art to use $n=2$, performed in triplicate (which is a standard measure), as*
*a small population of $n=2$ has only 1 degree of freedom for estimating the standard deviation and*
*results in very wide confidence interval. At the very least, the authors should replicate their*
*experiments to reach at least an n of 3 (as the authors have done it in Figure 1 panel (a)). Also, the*
*authors should use Standard deviation (as done in Figure 1a) throughout as it allows a better*
*judgement of the quality of the experimental data and their variability.*

We have repeated our experiments with an n of 3, each performed in triplicate, with standard
deviations included throughout.

*The figures which need to be adjusted would be Figures 1b, 2c, 3c (and “standard deviation from the*
*mean” is not a standard term), extended data figures 7 (again “standard deviation from the mean”).*

The figures have been adjusted.

*For better comparability, the authors should also include the uptake data of wildtype human DAT in*
*figure 1 a and b, and also the obtained Km, Vmax and Kd values in the respective figure legends.*

We have included the uptake data of the wild-type full length hDAT along with the Δ -hDAT
construct and provide Km and Vmax in Figure 1a.

For Figure 1b, the purpose of the experiment was to show that Δ -hDAT, purified in detergent
conditions in which we are preparing cryo-EM samples, retains robust ligand binding activity. For this
we separately purified Δ -hDAT for scintillation proximity assays using detergent and buffer conditions
similar to those employed in the cryo-EM studies. We have not done ligand binding experiment with
full length hDAT in similar detergent conditions since the purification conditions are optimized for
the Δ -hDAT construct. We have compared the binding affinity obtained in our SPA experiment with
the values reported in the published literature (lines 78-79).

Minor points:

*Abstract (line 23): “the mechanisms by which it is inhibited by small molecules and Zn²⁺ remain*
*unknown.” The statement is too strong and negates decades of research; it is simply an overstatement*
*that nothing is known – that was some 60 years back, when Hertting and Axelrod started to elucidate*
*basic mechanisms (papers in nature & science, 1960). The only thing which can be stated here is that*
*the novel structure adds to a more precise structural understanding.*

We have toned down this statement.

*Abstract (line 32): The structure and also the manuscript fails in explaining how zinc would inhibit*
*transport. It just shows where it binds – in the current cryo-EM snapshot. Please modify the text*
*accordingly.*

We have further analyzed the zinc site in the context of different conformational states of hSERT and
can now suggest that EL4 moves relative to EL2 upon the transition from outward to inward states
and thus formation of the zinc site, which binds EL4 to EL2, inhibits transport by restricting this
essential conformational change (Fig 3b). We believe our complete description of the Zn²⁺ site and its
likely role in specifically restricting the conformational movement of EL4 provides new insight into
how Zn²⁺ inhibits transport (lines 302-318, 329-334).

Line 63: From a pharmacological point of view, cocaine cannot be viewed as “acting with high affinity to block transport activity”. Please adjust – also, cocaine has similar affinities across monoamine transporters, on a broader scale.

We have revised the manuscript accordingly.

Line 141: “differential residue composition in subsite C can explain,” – “can explain” is too strong a statement here, as it is a speculation. Please replace by a respective wording.

We have rephrased as a speculative statement.

Line 143: pSERT needs to be introduced and referenced, the PDB-code is not enough; citation 47 is needed here.

We have revised the text as suggested.

Figure 3: The figure shows F213, not F214. Please correct.

We have removed this figure from the manuscript.

Extended Data Figure 7. The MRS7292 site in Δ -hDAT and comparison with allosteric sites 846 in hSERT. Please label representative amino acid side chains in panel (a).

Extended Data Figure 7a has been removed. The density of MRS7292 is instead shown in Main Figure 2c.

Referee #3 (Remarks to the Author)

1. I suggest that the authors conduct MD simulations to validate the binding poses of the two ligands.

We have carried out substantial MD studies that are discussed in the manuscript (Figure 2f and Extended Data Fig. 6d, and lines 160-171 and 246-261).

2. Since the authors did not obtain the structure of the allosteric inhibitor MRS7292 bound to DAT, how can they further prove that MRS7292 inhibits dopamine transport? Could MD simulations or other experiments help explain this mechanism?

We have elucidated the structure of Δ -hDAT in complex with the allosteric ligand MRS7292 and the current reconstruction of Δ -hDAT allowed for unambiguous assignment of MRS7292 molecule to the associated density. MRS7292 mediated non-competitive inhibition of dopamine transport has been

shown in a previous study (Tosh et al., 2017; PMID: 28319392) and also by our group (Navratna et.al.,
2018; PMID: 29965988). In this manuscript, we have done extensive molecular characterization of the
MRS7292 and Δ -hDAT interactions via mutations and dopamine inhibition transport studies (Fig. 2d,
Extended Data Fig. 7c). We have also studied the interactions via MD simulation, as described in the
main text and in the Supplement.

*3. Would there be a synergistic effect when using MRS7292 in conjunction with the β -CFT central*
*inhibitor? Does MRS7292 exhibit dose dependency for central site ligand binding?*

Yes, MRS7292 non-competitively blocks dopamine uptake and enhances binding of β -CFT at the
central site (Navratna et al., 2018). MRS7292 has been found to enhance binding of the related
phenyltropane based central site radioligand (RTI-55) in a dose dependent manner (Tosh et. al.,
2017).

*4. Since MRS7292 exhibits selectivity for hDAT, have the mutation experiments*
*demonstrated how this ligand achieves high specificity for DAT as compared to hNET and hSERT?*

We appreciate this comment and have carried out extensive experiments to probe the MRS binding
site, as described in the main text ('MRS7292 sculpts an allosteric binding site', Fig. 2d and Extended
Data Fig. 7c). In summary, we have found that while several specific interactions are associated with
potent inhibition of Δ -hDAT uptake by MRS7292, the consequences of other substitutions suggest that
the specificity of Δ -hDAT for the MRS7292 compound is distributed amongst residues that are not
directly in contact with MRS. For example, mutation of D385 to alanine, the equivalent residue found
in hSERT, resulted in nearly 2 fold increase in the IC₅₀ value, yet substitution with an asparagine had
little effect. These mutants thus suggest that a hydrogen bond interaction between the side chain of
D385 and the N1 nitrogen of MRS7292, perhaps mediated by a water molecule, specifically bolsters
binding of MRS7292 to hDAT. By contrast, Δ -hDAT residues V364 and I390, which are isoleucine and
leucine in hSERT, respectively, when mutated to their hSERT counterparts in Δ -hDAT, actually
enhance the ability of MRS7292 to inhibit hDAT uptake activity. These results thus suggest that
determinants other than direct contacts between the residues in the MRS7292 binding site and the
MRS7292 ligand can play a role in conferring MRS7292-mediated inhibition of uptake activity. Further
extensive studies, which are beyond the scope of the present work, will be required to elaborate all of
the determinants of MRS7292 specificity.

*5. There should be an FSC_{work}/FSC_{free} curve to demonstrate the absence of overfitting in model*
*refinement. A model versus map curve should also be included.*

We have included an FSC_{work}/FSC_{free} curve in Extended Data Fig 2e. The marginal difference
between the FSC_{work} and FSC_{free} curves indicates there is no significant overfitting in the model
refinement.

*6. Given that hDAT is not very large, could the author describe cryo-EM techniques briefly in the*
*main text?*
We have added a brief description in the main text (lines 83-87).
*7. Please include the densities of both ligands in the figures in the main text.*
We have included the density for β -CFT in main Figure 1f and for MRS7292 in Figure 2c.
*8. Show the key residues in Extended Data Figure 7e.*
We have shown the key residues in the MRS7292 binding pocket in Extended Data Figure 7e (Figure
7f in the revised manuscript).
*9. In Extended Data Table 2, the percentage of poor rotamers is somewhat high. Please try refining*
*the structure again.*
The final statistics of the latest Δ -hDAT model refined against the current improved map contains no
rotamer outliers. The Extended Data Table has been updated accordingly.
*10. Line 112-113, the glycosylation of residues on EL2 is proposed to be related with conformation*
*mobility, it is interesting point, could the author provide more evidence to prove that?*
We have toned down this comment.

Reviewer Reports on the First Revision:

Referees' comments:

Referee #1 (Remarks to the Author):

The manuscript has significantly improved. In particular the improved cryoEM reconstruction significantly strengthens the claims in the manuscript. I only have a few minor suggestions:

Line 274 - 294: This section focuses on the aspect of why the hSERT S2 site is NOT present in hDAT, which could have been achieved by AlphaFold predictions and sequence alignments, and I do not believe warrants this extensive of a discussion.

I am, however, curious, whether the authors believe that the MRS7292-site overlaps with the second binding site of dopamine on hDAT, analogous to 5-HT at SERT? Or in a more general sense: are there any other ligand, that would likely bind to the MRS7292-site?

Line 296: "Vesicle fusion" suggests fusion of multiple vesicles. Rephrase.

Line 308: It is difficult to understand the logic behind the argument of 4 μM Zn^{2+} being present from cell lysis. The sample has undergone >24 hour purification protocols with extensive washes with buffers that do not contain Zn.

The authors discuss Zn^{2+} being a unique inhibitor among SLC6 family members, there is some literature on certain GABA transporter subtypes being regulated by Zn^{2+} in a similar fashion to hDAT (PMID: 15829583).

Referee #2 (Remarks to the Author):

The manuscript by Srivastava and colleagues has been greatly improved and most of the requested additions and edits have been carried out. The additional data with respect to the allosteric modulatory compound are interesting and support the structure.

There are only a few remaining issues that I would like to clarify. One important issue that is still puzzling to me is that the N-terminally truncated DAT is even showing higher V_{max} values compared to the non-truncated wildtype DAT – which is in stark contrast to previous data from several groups. The data however speak for themselves and will remain puzzling (probably not only for me) – and one would probably need to examine the I248Y variant, whether this one adds so much to the transport capability of the truncated transporter.

Line 89: "Both in the single particle classifications, and in the biochemical analysis of the transporter, we observe delta-hDAT as a monomer..." – here, the authors would need to add "detergent-solubilized"

after 'observe' to make clear why this may be the case. It could also be good to add that many of the numerous studies have been performed in living cells devoid of any of the harsh treatments that have been utilized in the present work. I would even go so far to think about deleting or at least adapting the subsequent sentence as it has been shown both by the groups of Javitch and also Sorkin that detergent solubilization can preserve oligomeric assembly of DAT. However, this is certainly not the point of the manuscript.

The data in Figure 3c (Zinc-induced inhibition of DAT) are much better than before, but still they are not really resembling what had been published before by Norregaard & Stockner. Again, this is not the main point of the exercise here – and the authors can at least show that they somewhat reproduce the effect. The high and low affinity inhibition is not reproduced at all and hence, I would simply say that this is an average data set.

Regarding the MD simulations displayed in Figures 2f und SI-Figures 6d – where are the other replicas – please show them for the sake of completeness.

Minor:

I stumbled over the use of the marketed names of Ritalin and Adderall and I do not know why it is necessary to use them – they are only valid in on area of the planet. The generic names are more important because they are valid everywhere. In addition, the mode of action of methylphenidate and “amphetamines” is completely different in that the first blocks the transporter while the second induces transporter-mediated efflux. This would probably be more relevant to be added – the authors have also described the functional interaction between cocaine and DA-transporter.

Referee #3 (Remarks to the Author):

The author has addressed my concerns and included additional data into the revised manuscript. The current version of ms was well written, I recommend to publish this interesting paper.

Author Rebuttals to First Revision:

Referees' comments:

Referee #1 (Remarks to the Author):

The manuscript has significantly improved. In particular the improved cryoEM reconstruction significantly strengthens the claims in the manuscript. I only have a few minor suggestions:

Comment. *Line 274 - 294: This section focuses on the aspect of why the hSERT S2 site is NOT present in hDAT, which could have been achieved by AlphaFold predictions and sequence alignments, and I do not believe warrants this extensive of a discussion.*

Reply. While we understand this comment, we do believe that a discussion of the hDAT equivalent of the hSERT S2 site is relevant, given the importance of the S2 site in hSERT for the binding of therapeutic drugs and possibly also for hSERT translocation mechanism.

Comment. *I am, however, curious, whether the authors believe that the MRS7292-site overlaps with the second binding site of dopamine on hDAT, analogous to 5-HT at SERT? Or in a more general sense: are there any other ligand, that would likely bind to the MRS7292-site?*

Reply. We also appreciate this comment. The MRS site does not overlap with the hDAT equivalent of the hSERT S2 site. Also, we do not know if there is any other ligand, such as an endogenous compound, that binds to the MRS site. Clearly, this is an interesting future avenue of research.

Comment. *Line 296: "Vesicle fusion" suggests fusion of multiple vesicles. Rephrase.*

Reply. Rephrased to state that the vesicle fusion is with the presynaptic membrane.

Comment. *Line 308: It is difficult to understand the logic behind the argument of 4 μ M Zn^{2+} being present from cell lysis. The sample has undergone >24 hour purification protocols with extensive washes with buffers that do not contain Zn.*

Reply. In this work, and in past work (PMID: 30500536), we have carefully and thoroughly documented the copurification of micromolar concentrations of zinc with eukaryotic membrane proteins.

Comment. *The authors discuss Zn^{2+} being a unique inhibitor among SLC6 family members, there is some literature on certain GABA transporter subtypes being regulated by Zn^{2+} in a similar fashion to hDAT (PMDID: 15829583).*

Reply. We appreciate this comment yet note we did not describe the activity of zinc on hDAT as 'unique' and we also note that while the above referenced paper from 2005 does describe the inhibition of GAT4 by micromolar concentrations of zinc, neither the residues nor the molecular mechanism by which zinc acts on the transporter has since been well described. Given space and reference constraints, we have chosen to discuss and reference work where zinc action is more well defined and described.

Referee #2 (Remarks to the Author):

The manuscript by Srivastava and colleagues has been greatly improved and most of the requested additions and edits have been carried out. The additional data with respect to the allosteric modulatory compound are interesting and support the structure. There are only a few remaining issues that I would like to clarify.

Comment. *One important issue that is still puzzling to me is that the N-terminally truncated DAT is even showing higher Vmax values compared to the non-truncated wildtype DAT – which is in stark contrast to previous data from several groups. The data however speak for themselves and will remain puzzling (probably not only for me) – and one would probably need to examine the I248Y variant, whether this one adds so much to the transport capability of the truncated transporter.*

Reply. We appreciate this comment yet note that we are measuring transport in the context of cells and have not normalized the data for transporter expression levels, something that should be done in order to make comparisons between different constructs. We further note that the Km value for Δ -hDAT is similar to the full length transporter.

Line 89: “Both in the single particle classifications, and in the biochemical analysis of the transporter, we observe delta-hDAT as a monomer...” – here, the authors would need to add “detergent-solubilized” after ‘observe’ to make clear why this may be the case. It could also be good to add that many of the numerous studies have been performed in living cells devoid of any of the harsh treatments that have been utilized in the present work. I would even go so far to think about deleting or at least adapting the subsequent sentence as it has been shown both by the groups of Javitch and also Sorkin that detergent solubilization can preserve oligomeric assembly of DAT. However, this is certainly not the point of the manuscript.

Reply. We have added 'detergent-solubilized' as suggested.

Comment. *The data in Figure 3c (Zinc-induced inhibition of DAT) are much better than before, but still they are not really resembling what had been published before by Norregaard & Stockner. Again, this is not the main point of the exercise here – and the*

authors can at least show that they somewhat reproduce the effect. The high and low affinity inhibition is not reproduced at all and hence, I would simply say that this is an average data set.

Reply. We agree with the reviewer that the zinc inhibition is not the main point of the manuscript and also emphasize that multiple authors, including Norregaard, noted that the high concentrations of zinc used to explore the low affinity inhibition may also simply be toxic to the cells, thus reducing substrate uptake. Because only the high affinity inhibition is relevant to our study, we focused on lower concentrations of zinc.

Comment. *Regarding the MD simulations displayed in Figures 2f und SI-Figures 6d – where are the other replicas – please show them for the sake of completeness.*

Reply. We have included all the replicas in Figure 2f and Extended Data Figure 6d. We have previously supplied the requested replicas in the revised manuscript, in Supplementary Information Figures 5 and 8 (6 and 9 in the current version).

Minor:

Comment. *I stumbled over the use of the marketed names of Ritalin and Adderall and I do not know why it is necessary to use them – they are only valid in on area of the planet. The generic names are more important because they are valid everywhere. In addition, the mode of action of methylphenidate and “amphetamines” is completely different in that the first blocks the transporter while the second induces transporter-mediated efflux. This would probably be more relevant to be added – the authors have also described the functional interaction between cocaine and DA-transporter.*

Reply. We have replaced Ritalin and Adderall with methylphenidate and amphetamines, respectively.

Referee #3 (Remarks to the Author):

The author has addressed my concerns and included additional data into the revised manuscript. The current version of ms was well written, I recommend to publish this interesting paper.